# Structural basis for mammalian nucleotide sugar transport

Shivani Ahuja, Matthew R Whorton*

Vollum Institute, Oregon Health & Science University, Portland, United States

**Abstract** Nucleotide-sugar transporters (NSTs) are critical components of the cellular glycosylation machinery. They transport nucleotide-sugar conjugates into the Golgi lumen, where they are used for the glycosylation of proteins and lipids, and they then subsequently transport the nucleotide monophosphate byproduct back to the cytoplasm. Dysregulation of human NSTs causes several debilitating diseases, and NSTs are virulence factors for many pathogens. Here we present the first crystal structures of a mammalian NST, the mouse CMP-sialic acid transporter (mCST), in complex with its physiological substrates CMP and CMP-sialic acid. Detailed visualization of extensive protein-substrate interactions explains the mechanisms governing substrate selectivity. Further structural analysis of mCST's unique lumen-facing partially-occluded conformation, coupled with the characterization of substrate-induced quenching of mCST's intrinsic tryptophan fluorescence, reveals the concerted conformational transitions that occur during substrate transport. These results provide a framework for understanding the effects of disease-causing mutations and the mechanisms of this diverse family of transporters.
DOI: https://doi.org/10.7554/eLife.45221.001

## Introduction

Nucleotide-sugar transporters (NSTs) are products of the solute carrier 35 (SLC35) gene family in humans and are a critical part of the glycosylation machinery in all eukaryotes (*Hadley et al., 2014*; *Ishida and Kawakita, 2004*; *Song, 2013*). They are responsible for transporting nucleotide sugars from the cytoplasm, where they are synthesized, into the Golgi lumen where they are then utilized by glycosyltransferases to glycosylate proteins and lipids (*Figure 1—figure supplement 1A*) (*Capasso and Hirschberg, 1984*; *Milla and Hirschberg, 1989*; *Tiralongo et al., 2006*; *Waldman and Rudnick, 1990*). NSTs provide an additional essential role of transporting the nucleotide monophosphate (NMP) byproduct of the glycosyltransferase reaction back to the cytoplasm where it can be recycled. Since many glycosyltransferases are inhibited by NMPs, this antiport property of NSTs provides an additional layer of regulation of glycan synthesis (*Hirschberg et al., 1998*). Many NSTs are not obligatory antiporters, but the presence of NMPs in the lumen accelerates nucleotide sugar uptake several-fold through a poorly-understood process termed 'trans-stimulation' (*Capasso and Hirschberg, 1984*; *Milla and Hirschberg, 1989*; *Tiralongo et al., 2006*; *Waldman and Rudnick, 1990*).

Glycosylation is the most common form of protein and lipid modification. It affects nearly every aspect of biology by modifying protein folding, stability, and functional properties (*Dwek et al., 2002*; *Moremen et al., 2012*; *Ohtsubo and Marth, 2006*; *Stanley, 2011*). Glycans are typically terminated by sialic acids, which is important for many cell adhesion processes but also serves as ligands for pathogen invasion (*Varki, 2008*; *Varki and Schauer, 2009*). Since NST activity controls the concentrations of nucleotide sugars in the Golgi lumen, disruption of NST activity can have many adverse physiological effects. There are a number of genetic diseases caused by mutations in human NST genes that have severe and debilitating phenotypes (*Jaeken and Matthijs, 2007*; *Song, 2013*). In addition, some NSTs are potently inhibited by antiviral nucleoside analogs, which may form the

*For correspondence:
whorton@ohsu.edu

Competing interests: The authors declare that no competing interests exist.

**eLife digest** The cells in our body are tiny machines which, amongst other things, produce proteins. One of the production steps involves a compartment in the cell called the Golgi, where proteins are tagged and packaged before being sent to their final destination. In particular, sugars can be added onto an immature protein to help to fold it, stabilize it, and to affect how it works.

Before sugars can be attached to a protein, they need to be 'activated' outside of the Golgi by attaching to a small molecule known as a nucleotide. Then, these 'nucleotide-sugars' are ferried across the Golgi membrane and inside the compartment by nucleotide-sugar transporters, or NSTs. Humans have seven different kinds of NSTs, each responsible for helping specific types of nucleotide-sugars cross the Golgi membrane. Changes in NSTs are linked to several human diseases, including certain types of epilepsy; these proteins are also important for dangerous microbes to be able to infect cells. Yet, scientists know very little about how the transporters recognize their cargo, and how they transport it.

To shed light on these questions, Ahuja and Whorton set to uncover for the first time the 3D structure of a mammalian NST using a method known as X-ray crystallography. This revealed how nearly every component of this transporter is arranged when the protein is bound to two different molecules: a specific nucleotide, or a type of nucleotide-sugar. The results help to understand how changes in certain components of the NST can lead to a problem in the way the protein works. Ultimately, this knowledge may be useful to prevent diseases linked to faulty NSTs, or to stop microbes from using the transporters to their own advantage.

DOI: https://doi.org/10.7554/eLife.45221.002

basis for some of the side effects associated with this class of drugs (*Chiaramonte et al., 2001*; *Hall et al., 1994*). On the other hand, NSTs also represent potential therapeutic targets, as NSTs are virulence factors in many types of parasites and fungi that require extensive cell-surface glycoconjugates for effective pathogenesis (*Caffaro et al., 2013*; *Descoteaux et al., 1995*; *Engel et al., 2009*; *Hong et al., 2000*; *Liu et al., 2013*; *Ma et al., 1997*). Additionally, altered glycosylation profiles of cell-surface proteins are a property of malignant cells, and studies have shown that pharmacological block of some NSTs can inhibit tumor metastasis (*Caffaro and Hirschberg, 2006*; *Esko and Bertozzi, 2009*; *Hadley et al., 2014*; *Ohtsubo and Marth, 2006*; *Song, 2013*; *Stowell et al., 2015*; *Wang et al., 2016*).

Mammals express seven NSTs, which collectively transport the eight nucleotide sugars used in Golgi glycan synthesis (*Figure 1—figure supplement 1B*). Each NST is highly selective for either one or two nucleotide sugars and the corresponding NMP – a critical property that is required for the proper hierarchical assembly of glycans. However, little is known about the underlying structural basis for this specificity or the mechanisms of substrate antiport. Here, we report X-ray crystal structures of the mouse CMP-sialic acid (CMP-Sia) transporter (mCST) in complex with CMP and CMP-Sia at 2.6 Å and 2.8 Å resolution, respectively. Analysis of these structures, coupled with functional characterization of mCST, elucidates the fundamental principles of the structure-function relationship of NSTs.

## Results and discussion

### Construct characterization and structure determination

We identified mCST, which is 91% identical to human CST (hCST; *Figure 1—figure supplement 2*), as a suitable candidate for structural studies after screening a large panel of NST orthologs. One obstacle that we encountered when characterizing this construct was our discovery that commercial stocks of CMP-Sia contain approximately 10% CMP (*Figure 1—figure supplement 3A*), which has a ~ 100 fold higher affinity towards mCST (*Figure 1A*). In addition, CMP-Sia hydrolyzes in aqueous solutions to yield CMP and Sia (*Beau et al., 1984*; *Horenstein and Bruner, 1996*) (*Figure 1—figure supplement 3B–H*). To our knowledge, these issues have not been acknowledged or addressed in the CST literature. Therefore, we developed a method where we treat CMP-Sia with a nonselective phosphatase, Antarctic phosphatase (AnP), to convert all of the free CMP to cytidine (*Figure 1—*

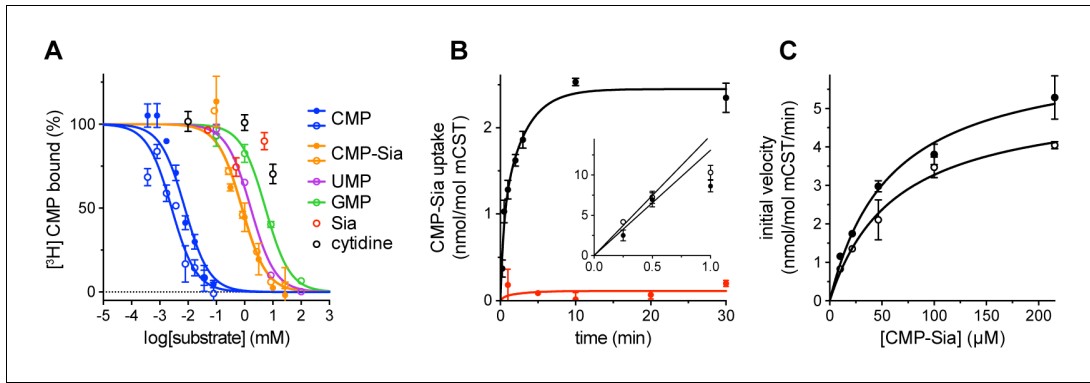

**Figure 1.** Functional properties of mCST. (**A**) Purified, detergent-solubilized mCST was bound to SPA beads and the indicated compound was titrated against 30 nM [$^3$H]CMP to determine its affinity. Solid symbols denote titrations done with mCST and the open symbols are for mCSTΔC. (**B**) Time course of 30 μM CMP-Sia uptake for purified mCST reconstituted into lipid vesicles with (black) or without (red) 300 μM CMP inside the vesicles. (Inset), The first minute of CMP-Sia uptake for either mCST (solid circles) or mCSTΔC (open circles) reconstituted into CMP-filled vesicles is shown to illustrate the linear relationship between uptake and time for up to 30 s, which we used to determine the initial velocity of transport for a given concentration of CMP-Sia. (**C**) The initial velocity of CMP-Sia uptake for either mCST (solid circles) or mCSTΔC (open circles) reconstituted into CMP-filled vesicles is plotted as a function of substrate concentration to determine the $K_m$ and $V_{max}$ for transport. In all panels, the symbols show the mean ± standard error of the mean (SEM) for $n = 2$, except for the CMP titrations in (**A**), and the titration for mCST in (**C**) where $n = 4$.

DOI: https://doi.org/10.7554/eLife.45221.003

The following figure supplements are available for figure 1:

**Figure supplement 1.** Description of SLC35 proteins.
DOI: https://doi.org/10.7554/eLife.45221.004
**Figure supplement 2.** Sequence alignment of mCST with other SLC35 and DMT proteins.
DOI: https://doi.org/10.7554/eLife.45221.005
**Figure supplement 3.** Detection and elimination of CMP contaminants in CMP-Sia stocks.
DOI: https://doi.org/10.7554/eLife.45221.006

**Table 1.** Functional properties of various mCST constructs.

The $K_d$ for the indicated ligand was determined by either: 1) a full SPA titration as shown in **Figure 1A** 2) fitting the two-point SPA data from **Figure 5H** to get a crude estimation of the $K_d$, or 3) fitting the tryptophan fluorescence quenching data shown in **Figure 5G** ($K_d$'s were not fit for the Trp207 mutants which showed no quenching). The $K_m$ and $V_{max}$ were determined from data shown in **Figure 1C**.

| Construct | Ligand | $K_d$ (μM) | | | $K_m$ (μM) | $V_{max}$ (nmol/mol mCST/min) |
| | | SPA titration | SPA 2-point | Trp-FL | | |
|---|---|---|---|---|---|---|
| mCST | CMP | 6.3 ± 1.2 | 5.5 ± 1.6 | 2.9 ± 1.1 | | |
| | CMP-Sia | 482 ± 147 | | | 58.1 ± 13.2 | 6.5 ± 0.6 |
| mCSTΔC | CMP | 1.8 ± 0.4 | | 11.1 ± 2.9 | | |
| | CMP-Sia | 295 ± 100 | | | 62.0 ± 14.7 | 5.3 ± 0.5 |
| | UMP | 1621 ± 1119 | | | | |
| | GMP | 5376 ± 1151 | | | | |
| mCST (W145L) | CMP | | 0.8 ± 0.3 | 3.3 ± 1.0 | | |
| mCST (W160L) | CMP | | 1.6 ± 0.4 | 11.9 ± 2.4 | | |
| mCST (W207L) | CMP | | 1.5 ± 0.4 | N/D | | |
| mCST (W207F) | CMP | | 3.8 ± 1.5 | N/D | | |
| mCST (W247L) | CMP | | 2.4 ± 1.5 | 7.4 ± 2.6 | | |

DOI: https://doi.org/10.7554/eLife.45221.011

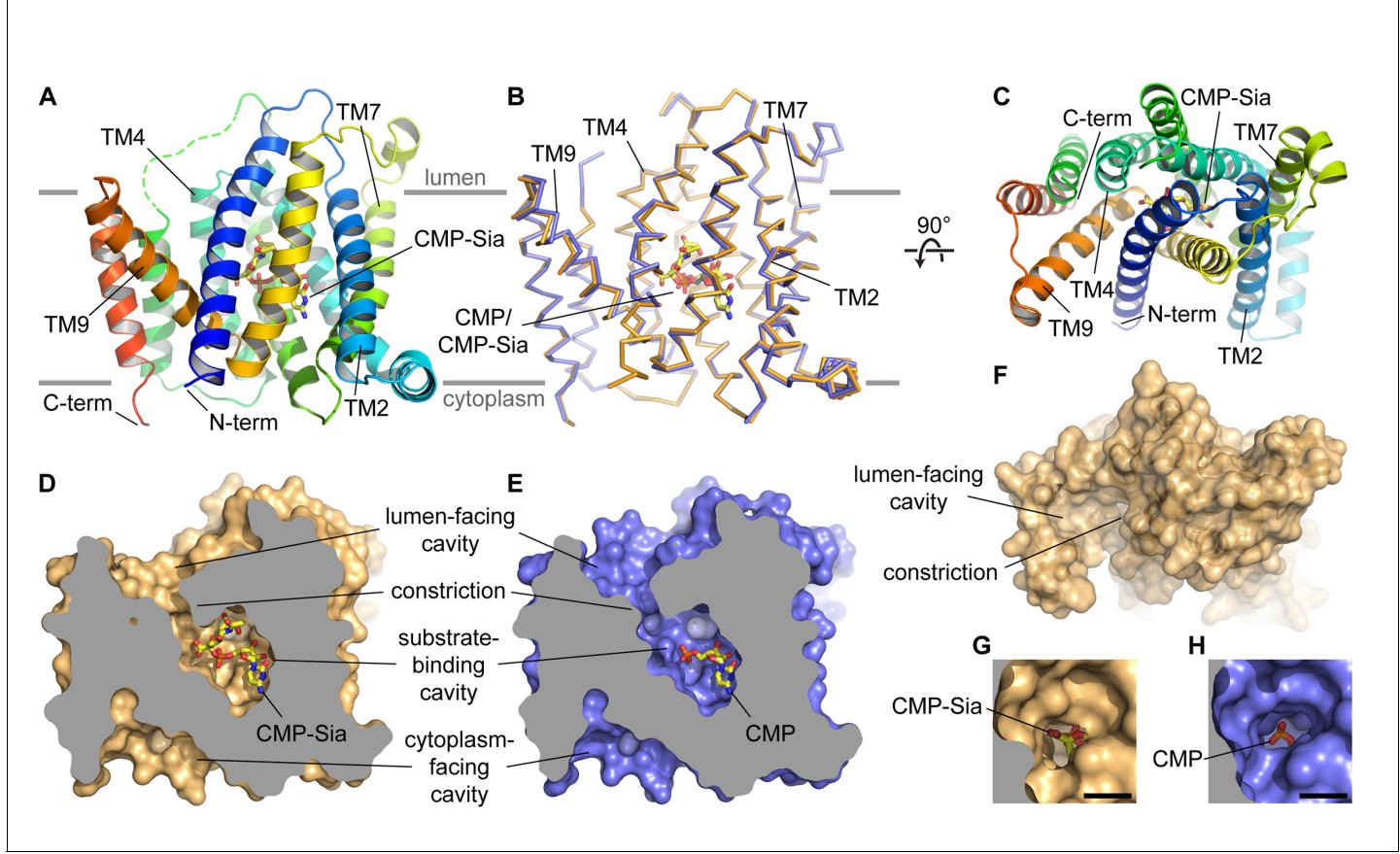

**Figure 2.** Overall structure of mCST. (**A and B**) Side views of mCST are shown. CMP or CMP-Sia is shown as sticks with carbon atoms colored yellow. The approximate boundary of the lipid bilayer is indicated by the horizontal gray lines. In (**A**), only the mCST-CMP-Sia structure is shown and the dashed green line indicates the connectivity of the disordered linker between TMs 5 and 6. In (**B**), the mCST-CMP-Sia (orange) and mCST-CMP (blue) structures are superimposed. (**C**) Top-down view of the mCST-CMP-Sia structure from the lumenal side. The protein is colored the same as in (**A**). (**D and E**) The mCST-CMP-Sia (**D**, orange) and mCST-CMP (**E**, blue) structures are shown in surface representation with the front half of the protein sliced away to reveal the shape of the substrate binding cavities. (**F**) A surface representation of a top-down view (same as in **C**) of the mCST-CMP-Sia structure is shown to highlight the shape of the lumen-facing cavity and the location of the constriction. (**G and H**) A close-up top-down view of the constriction from the lumen-facing cavity is shown for the mCST-CMP-Sia (**G**, orange) and mCST-CMP (**H**, blue) structures. The black scale bar in each is 5 Å.

DOI: https://doi.org/10.7554/eLife.45221.007

The following figure supplements are available for figure 2:

**Figure supplement 1.** Electron density maps.

DOI: https://doi.org/10.7554/eLife.45221.008

**Figure supplement 2.** Comparison of the mCST structures and crystal lattices.

DOI: https://doi.org/10.7554/eLife.45221.009

**Figure supplement 3.** Pseudo-symmetry in mCST.

DOI: https://doi.org/10.7554/eLife.45221.010

*figure supplement 3B*), which does not bind mCST (*Figure 1A*). For binding and transport experiments, the AnP is removed by ultrafiltration and we discuss in the Methods how we account for any CMP-Sia hydrolysis that occurs during these experiments.

When mCST is purified in detergent, it retains saturable substrate binding, with a $K_d$ of $6.3 \pm 1.2$ µM and $482 \pm 147$ µM for CMP and CMP-Sia, respectively (*Figure 1A*). Purified mCST also transports CMP-Sia when reconstituted into liposomes, with a $V_{max}$ of $6.5 \pm 0.6$ nmol/mol mCST/min and a $K_m$ of $58.1 \pm 13.2$ µM (*Figure 1B and C*), which is within the range of reported $K_m$'s for CMP-Sia transport for human or mouse CST expressed in Golgi (*Aoki et al., 2003*; *Chiaramonte et al., 2001*; *Milla and Hirschberg, 1989*; *Tiralongo et al., 2006*). CMP-Sia transport activity is greatly

**Table 2.** Data collection and refinement statistics.

| | HDVD mCST-CMP | | | LCP mCSTΔC | |
|---|---|---|---|---|---|
| | **Native** | **Hg** | **Pt** | **CMP** | **CMP-Sia** |
| **Data collection** | | | | | |
| Space group | $P2_1$ | $P2_1$ | $P2_1$ | C2 | C2 |
| Crystals (#) | 5 | 1 | 1 | 26 | 1 |
| Cell dimensions (Å) | | | | | |
| a | 51.82 | 51.88 | 51.50 | 50.15 | 50.42 |
| b | 193.96 | 194.00 | 193.46 | 49.53 | 50.12 |
| c | 66.44 | 66.89 | 66.31 | 137.9 | 132.33 |
| $\alpha$, $\gamma$ = 90; $\beta$ = (°) | 101.79 | 102.16 | 101.63 | 92.65 | 91.83 |
| Resolution (Å) | 49–3.4 × 3.4 × 4.6[*] (3.47–3.38) | 49–3.5 × 3.5 × 5.6[†] (3.59–3.50) | 49–4.2 × 4.2 × 7.6[†] (4.31–4.20) | 47–2.58 (2.65–2.58) | 48–2.75 (2.83–2.75) |
| $R_{merge}$ | 0.07 (0.95) | 0.14 (4.03) | 0.23 (2.62) | 0.14 (0.47) | 0.13 (1.37) |
| $R_{pim}$ | 0.03 (0.67) | 0.09 (2.43) | 0.14 (1.63) | 0.05 (0.28) | 0.07 (0.75) |
| $I / \sigma I$ | 6.7 (1.6) | 6.8 (0.5) | 4.9 (0.7) | 9.3 (2.2) | 6.0 (1.0) |
| $CC_{1/2}$ in outer shell | 0.65 | 0.20 | 0.30 | 0.79 | 0.41 |
| Completeness (%) | 67.3 (4.0)[*] | 99.0 (99.6) | 99.9 (100) | 97.8 (79.1)[‡] | 99.8 (99.4) |
| Redundancy | 3.6 (2.2) | 7.1 (7.4) | 6.8 (7.0) | 7.2 (3.3) | 4.1 (4.3) |
| **MIRAS phasing[§]** | | | | | |
| Phasing power (iso/ano) | | 2.39/1.16 | 1.47/0.29 | | |
| $R_{cullis}$ (iso/ano) | | 0.45/0.44 | 0.65/1.00 | | |
| Figure of merit (SHARP) | | 0.31 | | | |
| Figure of merit (DM) | | 0.89 | | | |
| **Refinement** | | | | | |
| Resolution (Å) | 49–3.4 (3.50–3.38) | | | 47–2.58 (2.67–2.58) | 48–2.75 (2.85–2.75) |
| No. reflections (No. in free set) | 11456 (573) | | | 10101 (514) | 8305 (404) |
| $R_{work}/R_{free}$ | 28.9/32.1[¶] (64.9/63.0)[**] | | | 24.1/25.2 (26.3/33.6) | 25.6/27.9 (37.5/37.8) |
| No. atoms | | | | | |
| Protein | 4502 | | | 2235 | 2244 |
| Ligand | 42 | | | 21 | 41 |
| Water | 0 | | | 11 | 3 |
| B-factors | | | | | |
| Protein | 149.4 | | | 47.8 | 81.1 |
| Ligand | 122.1 | | | 39.4 | 90.6 |
| Water | — | | | 42.8 | 58.9 |
| R.m.s. deviations | | | | | |
| Bond lengths (Å) | 0.007 | | | 0.008 | 0.007 |
| Bond angles (°) | 1.011 | | | 1.127 | 1.031 |

[*]The diffraction data are anisotropic. For phasing and model refinement, the data were anisotropically truncated and B-factor sharpened as described in the Methods section. The data collection and refinement statistics reflect this. The dataset was overall 94.7% (81.7% in high-resolution shell) complete before being truncated. For the truncated dataset, resolution shells up to 4.4 Å are at least 95% complete.

[†]The diffraction data are anisotropic; however, the datasets were left unmodified for phasing.

[‡]The lower completeness for the high-resolution shell is due to only a minority of the crystals diffracting to ~2.6 Å. Most of the crystals only diffracted to ~2.7 Å, as evidenced by the 2.72–2.65 Å shell being 98.8% complete.

[§]The acentric phasing power and $R_{cullis}$ for the isomorphous (iso) and anomalous (ano) signals are shown. Phase figure of merits are also shown after refinement in SHARP and after density modification with DM.

¶The refinement statistics in this column represent the model from the LCP mCSTΔC-CMP crystal being refined against the native HDVD mCST-CMP dataset.

**R-factors in the 3.81–3.64 Å and 4.26–4.01 Å shells, which are 34% and 79% complete, respectively, are 38.5/39.8% and 31.2/34.4% ($R_{work}$/$R_{free}$), respectively.

Values in parentheses are for the highest-resolution shell, unless otherwise indicated.

DOI: https://doi.org/10.7554/eLife.45221.012

reduced when CMP is not included inside the vesicles (*Figure 1B*), which is likely the result of the lack of trans-stimulation.

We initially obtained crystals of full-length mCST in complex with CMP using the hanging-drop vapor diffusion (HDVD) method, which diffracted anisotropically to a moderate resolution (3.4 × 3.4 × 4.6 Å). We calculated an experimental electron density map using multiple isomorphous replacement with anomalous scattering using Hg and Pt derivatives (*Figure 2—figure supplement 1A–B*). A model based on this map showed that the entire 20-residue C-terminus was disordered. After screening a series of deletion constructs, we identified one lacking the last 15 residues (termed mCSTΔC) that was functionally identical to the full-length protein (*Figure 1*, *Table 1*) and produced lipidic cubic phase (LCP) crystals that diffracted to 2.6 Å. We solved the structure of these crystals with molecular replacement using the model of full-length mCST as the search model, and built a detailed model of mCST that refined to R/R-free factors of 24.1/25.2% (*Figure 2*, *Figure 2—figure supplement 1C–D*, *Table 2*).

We grew LCP crystals of mCSTΔC in complex with CMP-Sia, which diffracted to 2.8 Å, by leaving AnP in CMP-Sia stocks and also adding additional AnP to maintain very low concentrations of CMP during crystal growth (*Figure 1—figure supplement 3C*). The structure was solved with molecular replacement using the mCSTΔC-CMP structure as a search model, and we refined the resulting model to R/R-free factors of 25.6/27.9% (*Figure 2*, *Figure 2—figure supplement 1E–H*, *Table 2*). The HDVD and LCP structures were overall very similar (*Figure 2—figure supplement 2A*), despite having very different crystal lattices (*Figure 2—figure supplement 2B–E*). This indicates that the crystal packing interactions had no impact on the overall conformation of mCST. We will only discuss the higher-resolution LCP structures in the rest of the text.

## Overall mCST architecture

The structure of mCST consists of a 10-TM bundle (*Figure 2A–2C*). The general topology resembles that seen in other members of the larger drug/metabolite transporter (DMT) superfamily, which includes proteins with 4–10 TMs (*Jack et al., 2001*; *Västermark et al., 2011*). The 10-TM DMT proteins are thought to have arisen from an ancient helix addition to the 4-TM proteins (typified by EmrE) followed by a gene duplication event. Although the two halves of mCST lack significant sequence identity, the 10-TM bundle of mCST is arranged with a pseudo-two-fold inverted symmetry (*Figure 2—figure supplement 3*), similar to what is seen in other 10-TM DMT protein structures (*Lee et al., 2017*; *Parker and Newstead, 2017*; *Tsuchiya et al., 2016*). The N- and C-termini are on the same side of the Golgi membrane and are thought to face the cytoplasm based on previous experiments using HA-tag insertions (*Eckhardt et al., 1999*).

We observe several intra-membrane interactions between mCST molecules in our various crystal lattices (*Figure 2—figure supplement 2B–E*). However, since we determined that hCST, which is biochemically identical to mCST, is a monomer when purified in detergent (*Figure 2—figure supplement 2F–G*), these interactions do not stem from a stable pre-formed oligomer that survived detergent extraction and purification. There is some evidence that suggests that some NSTs may form oligomers in the Golgi membrane (*Gao and Dean, 2000*; *Hong et al., 2000*; *Puglielli and Hirschberg, 1999*; *Puglielli et al., 1999*); however, no experiments have directly investigated the oligomeric status of CSTs. Further work will be needed to determine if the interactions we see in our crystal contacts are physiologically relevant.

The overall conformation of the CMP- and CMP-Sia-bound mCST structures is generally similar with a few key differences that will be discussed later (*Figure 2B*). CMP and CMP-Sia both bind a similarly-shaped cavity inside the transporter, which is roughly halfway between the cytoplasmic and lumenal sides of the protein (*Figure 2D and E*). An elliptically-shaped constriction (*Figure 2F–2H*) divides this cavity into two distinct regions: the substrate-binding cavity and the lumen-facing cavity.

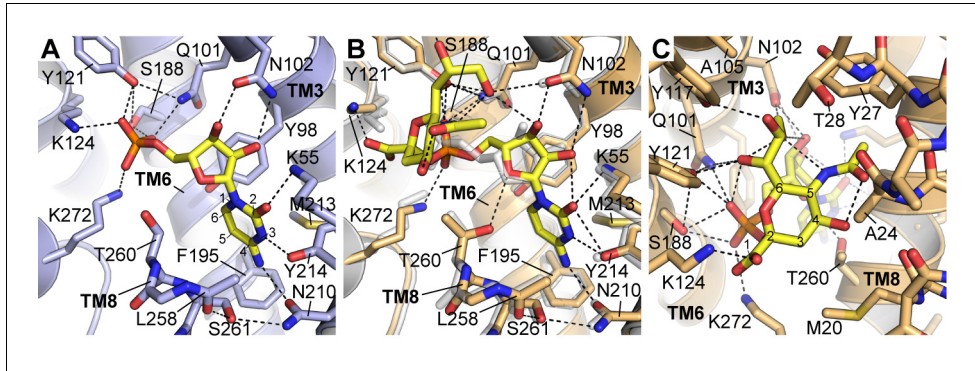

**Figure 3.** Details of CMP and CMP-Sia interactions with mCST. Either the mCST-CMP structure (**A**) or the mCST-CMP-Sia structure (**B, C**) are shown. (**B**) focuses on the CMP moiety of CMP-Sia whereas a different view is shown in (**C**) to better show the interactions with the Sia moiety. In all panels, key interacting residues are shown and dashed black lines indicate polar interactions. In (**A**) and (**B**), parts of TM8 are not shown for clarity. In (**B**), the mCST-CMP structure is superimposed (light gray) to show the different position of CMP and some of the interacting side chains. In (**A**) and (**C**), the ring atoms are numbered for cytosine (**A**) and Sia (**C**). In (**A**), the Thr260 hydroxyl interacts with the phosphate of CMP through a water, which is not shown (see *Figure 2—figure supplement 1I–J* for detailed views of the water interactions). In the mCST-CMP-Sia structure (**B**), the Sia moiety of CMP-Sia displaces this water, which causes Thr260 to flip and interact with the ribose of the CMP moiety.
DOI: https://doi.org/10.7554/eLife.45221.014

The following figure supplements are available for figure 3:

**Figure supplement 1.** Human diseases caused by SLC35 point mutations.
DOI: https://doi.org/10.7554/eLife.45221.015

**Figure supplement 2.** Homology modeling and docking of other nucleotide sugars and nucleotide analogs.
DOI: https://doi.org/10.7554/eLife.45221.016

The dimensions of this constriction (~6×4 Å) can perhaps permit the passage of CMP, but it is

**Table 3.** Human diseases caused by mutations in NST genes.

Only point mutations are shown; however, there are many examples of other types of mutations in SLC35 genes (e.g. insertions and deletions) that lead to frameshift or premature truncations that also cause disease (see *Edvardson et al., 2013*; *Hiraoka et al., 2007*; *Martinez-Duncker et al., 2005* and Online Mendelian Inheritance in Man (OMIM) database (https://omim.org)).

| Gene | Disease | Mutation | mCST residue | Biochemical phenotype | Possible mechanism based on the structure | Ref.[*] |
|---|---|---|---|---|---|---|
| SLC35A1 | Congenital disorder of glycosylation type IIf | Q101H | | CMP-Sia transport activity reduced by 50%. | Impairs substrate interaction. | 1 |
| SLC35A1 | Congenital disorder of glycosylation type IIf | T156R | | CMP-Sia transport activity reduced by 9-fold. | Impairs conformational transitions during transport. | 2 |
| SLC35A1 | Congenital disorder of glycosylation type IIf | E196K | | CMP-Sia transport activity reduced by 9-fold. | Impairs conformational transitions during transport. | 2 |
| SLC35A2 | Epileptic encephalopathy | S213F | S188 | No UDP-Gal transport observed. | Impairs substrate interaction. | 3 |
| SLC35A2 | Congenital disorder of glycosylation type IIm | V331I | V306 | UDP-Gal transport reduced by at least 60%. | Impairs conformational transitions during transport. | 4 |
| SLC35C1 | Congenital disorder of glycosylation type IIc | R147C | Y121 | Does not rescue fucosylation in cells lacking GDP-fucose transport. | Impairs substrate interaction. | 5, 6 |
| SLC35C1 | Congenital disorder of glycosylation type IIc | T308R | T284 | Does not rescue fucosylation in cells lacking GDP-fucose transport. | Impairs conformational transitions during transport. | 6, 7 |
| SLC35D1 | Schneckenbecken dysplasia | T65P | A38 | Near-complete loss of transport activity. | Impairs conformational transitions during transport. | 8 |

*References: 1 (*Mohamed et al., 2013*), 2 (*Ng et al., 2017*), 3 (*Kodera et al., 2013*), 4 (*Ng et al., 2013*), 5 (*Lühn et al., 2001*), 6 (*Lübke et al., 2001*), 7 (*Etzioni et al., 2002*), 8 (*Furuichi et al., 2009*)
DOI: https://doi.org/10.7554/eLife.45221.013

probably too small to let the bulkier Sia moiety of CMP-Sia pass through. For this reason, we think that this conformation of mCST represents a lumen-facing partially-occluded state. This state has not yet been observed in other 10-TM DMT protein structures and will be discussed in more detail below.

## Nucleotide substrate recognition and selectivity

Most members of the SLC35 family transport UMP and UDP-sugars, only one transports GMP and GDP-fucose, and only CST transports CMP and CMP-Sia (*Figure 1—figure supplement 1*). CST is highly selective for CMP, as UMP has a 20-fold higher $K_i$ for inhibiting CMP-Sia transport (*Chiaramonte et al., 2001*), and UMP and GMP have a 250- and 850-fold lower binding affinity than CMP, respectively (*Figure 1A* and *Table 1*). Our structure starts to explain how this is achieved.

In mCST, the various groups of CMP are extensively coordinated by at least fifteen residues (*Figure 3A*). Nearly a third of these interact with the phosphate group. The importance of these protein-phosphate interactions in mediating substrate binding is highlighted by: 1) the fact that cytidine alone does not bind mCST (*Figure 1A*), and 2) among all the residues that line the substrate-binding cavity, the only ones linked to known disease-causing mutations interact exclusively with the phosphate (*Figure 3—figure supplement 1* and *Table 3*). The cytosine group is closely coordinated by residues Lys55, Tyr214, and Asn210, which explains why mCST cannot easily accommodate the much larger guanine base of GMP. On the other hand, UMP is very similar to CMP; however, the structure suggests that there are two key differences that explain their differences in affinity. The first is that UMP has a carbonyl at the C-4 position of the uracil base instead of the amine that cytosine has (*Figure 1—figure supplement 1C*). In mCST, this amine is hydrogen-bonded to the Oδ1 of Asn210, with the orientation of Asn210's side chain stabilized by a hydrogen bond between the Nδ2 of Asn210 and the hydroxyl of Ser261 (*Figure 3A*). In this orientation, Asn210 will not be able to form a hydrogen bond with the C-4 carbonyl of uracil. The partial negative charge of uracil's carbonyl would also be electrostatically repelled by the partial negative charge of the π system of Phe195. The second difference between cytosine and uracil is that the nitrogen at position three is protonated in uracil. In mCST, the hydroxyl of Tyr214 is hydrogen-bonded to the N-3 of cytosine. If we assume that UMP would bind in a similar orientation as CMP, then the proton on the N-3 of uracil would be ~1.7 Å away from the hydroxyl oxygen of Tyr214. Therefore, it may still be able to hydrogen bond with Tyr214; however, NH–O hydrogen bonds are typically weaker than OH–N bonds hydrogen. In addition, tyrosine hydroxyls are generally thought to be relatively weak hydrogen bond acceptors on account of the lone pair of the hydroxyl's oxygen partially delocalizing within the adjacent aromatic ring (*McDonald and Thornton, 1994*).

These hypotheses are further supported by comparing the sequence of mCST to the closely-related UDP-Gal/GalNAc transporter (UGT, SLC35A2; 44% sequence identity) and the UDP-GlcNAc transporter (NGT, SLC35A3; 41% sequence identity) (*Figure 1—figure supplement 2*). Nearly all of the residues of mCST that are involved in coordinating CMP are conserved in UGT and NGT, with two key exceptions. The equivalent residues of Tyr214 and Ser261 in mCST are a Gly and Ala, respectively, in both UGT and NGT. An Ala at the Ser261 position would allow the conserved Asn at position 210 to orient its Nδ2 towards the binding pocket to be able to hydrogen bond with the C-4 carbonyl of uracil. On the other hand, the effect of having a Gly at the Tyr214 position is unclear. Sequence conservation in this region is high and homology models of UGT and NGT suggest that this Tyr-Gly substitution will result in the formation of a cavity that would be large enough to accommodate at least a few waters. Future work will be needed to understand what role, if any, a Gly at this position plays in affecting substrate specificity in UGT or NGT.

Although interactions between the protein and nucleobase are important for nucleotide selectivity, the additional extensive interactions between the protein and the rest of the nucleotide are apparently employed by some exogenous ligands to achieve high-affinity binding. This is seen with the antiviral nucleoside analog azidothymidine monophosphate (AZTMP). This molecule inhibits CST's CMP-Sia transport activity with a $K_i$ similar to that of CMP (*Chiaramonte et al., 2001*; *Hall et al., 1994*), despite the fact that the thymine base of AZTMP is essentially a methylated uracil, which would not be expected to have a strong interaction with CST. Instead, docking AZTMP into mCST suggests that the azide modification of the ribose group likely forms additional compensating interactions with the protein (*Figure 3—figure supplement 2A–B*), which could be exploited in future efforts to pharmacologically target NSTs.

## Nucleotide-sugar substrate recognition and selectivity

In the mCST-CMP-Sia structure, the Sia moiety of CMP-Sia occupies a large cavity that was present but vacant in the CMP-only structure (*Figure 2D and E*) whereas the CMP moiety of CMP-Sia binds in a similar but slightly different orientation (*Figure 3B*). The Sia moiety interacts with several residues of mCST through the various functional groups on the sugar ring (*Figure 3C*). The C-1 carboxyl and C-6 glycerol groups make several polar interactions, including an ionized hydrogen bond between the C-1 carboxyl and Lys124. This residue interacts with the α-phosphate of CMP in the CMP-only structure, but is shifted 2 Å to the side to accommodate Sia's C-1 carboxyl. Docking of UDP-sugars into homology models of UGT and NGT suggest that this conserved lysine in these transporters would interact with the β-phosphate of the UDP moiety (*Figure 3—figure supplement 2D–F*).

The C-4 hydroxyl and C-5 N-acetyl groups do not make any direct polar interactions with the protein but may interact with a number of residues through van der Waals forces. Cavities adjacent to the C-5 N-acetyl could likely accommodate one of the most common sialic acid derivatives where the C-5 substituent is an N-glycolyl instead of an N-acetyl, which are also referred to as Neu5Gc and Neu5Ac, respectively (*Figure 1—figure supplement 1C*, *Figure 3—figure supplement 2G–H*) (*Varki and Schauer, 2009*). Neu5Ac is converted to Neu5Gc by a cytoplasmic hydrolase that is found in most mammals except for humans. However, Neu5Gc is still found in human glycans, which is thought to arise primarily from dietary sources and has been associated with an increased risk for inflammation and carcinomas (*Samraj et al., 2014*; *Varki, 2008*; *Varki and Schauer, 2009*). The protein sequence for the region around the N-acetyl is nearly identical between mCST and hCST, which explains how hCST can still transport CMP-Neu5Gc.

Overall, the Sia moiety of CMP-Sia makes far fewer protein contacts than the CMP moiety. This difference is consistent with the observation that, unlike CMP, Sia alone does not bind mCST (*Figure 1A*) and does not competitively inhibit CMP-Sia transport (*Carey et al., 1980*; *Perez and Hirschberg, 1987*). This suggests that mCST's mechanism for discriminating among different nucleotides, as described above, is also the primary means for nucleotide-sugar selectivity. An additional component of selectivity may also arise from the large volume of the Sia-binding cavity, which may not be able to effectively coordinate the other smaller nucleotide-coupled sugars.

On the other hand, UDP-sugar transporters must utilize the converse of these principles in order to discriminate among the six UDP-coupled sugars. This concept is supported by sequence comparison between mCST and UGT/NGT (*Figure 1—figure supplement 2*), which shows that the most significant differences among the Sia-binding-site residues are the exchange of several non-polar and small side chains in CST to longer, mostly polar side chains in UGT and NGT. Homology models of UGT and NGT suggest this would result in a smaller binding pocket, which would clash with the various groups of Sia, but would be able to make several key interactions with substrates that may be important in substrate selectivity (*Figure 3—figure supplement 2C–F*).

## Structural basis for the differential binding affinities of CMP and CMP-Sia

Compared to the CMP-bound structure, the N-terminal two-thirds of TM1 of the CMP-Sia-bound structure twists and moves toward the substrate binding pocket (*Figures 2B* and *4A*). This brings three residues from TM1 (Met20, Ala24, and Tyr27) into close contact with the Sia moiety and is also associated with several other conformational changes in various regions of the protein.

The first is that there is a ∼ 0.5 Å displacement of the adjacent TM8 which is also associated with smaller but significant displacements of the other nearby TMs surrounding the CMP site (*Figure 4A*). These movements cause the side chains surrounding the CMP site to generally move away from each other, creating a slightly larger and differently-shaped binding pocket (*Figure 4B–4E*) – the latter of which is reflected in how the CMP moiety of CMP-Sia binds in a different orientation than CMP alone (*Figure 3B*). The second conformational difference associated with CMP-Sia binding is that TM1 is brought into closer contact with TM9 (*Figure 4F–G*). This is associated with a mostly rigid-body reorientation of a three-helix domain comprised of TMs 5, 9, and 10 (*Figure 4H*), which will be discussed in more detail in another section below. This increased interaction between TMs 1 and 9 is also associated with TM1 becoming generally more ordered. For example, we were able to see electron density for all of TM1, starting at Asn7, in the CMP-Sia-bound structure.

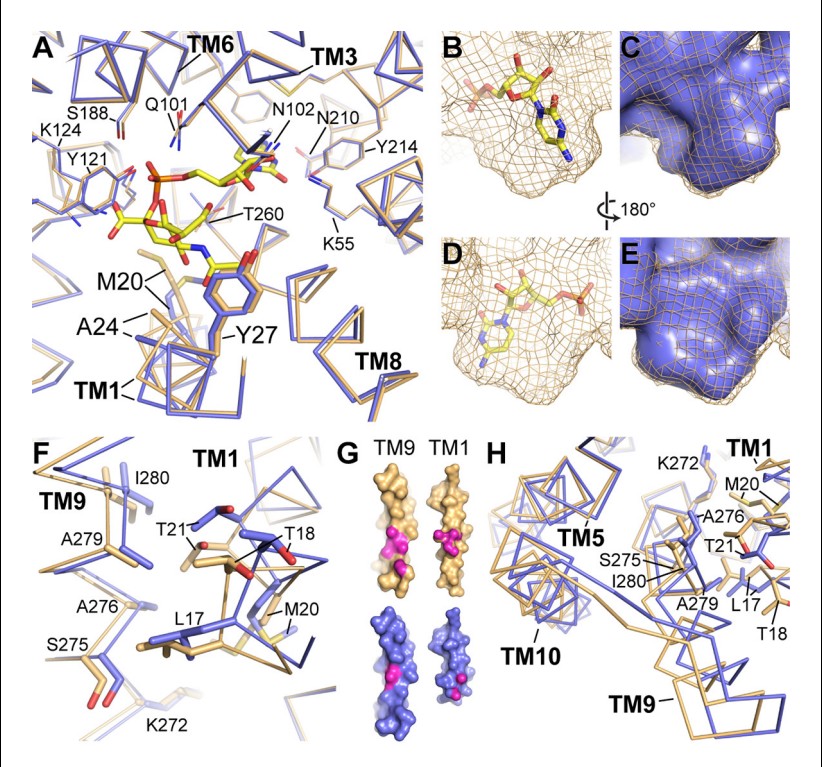

**Figure 4.** Differences between the CMP and CMP-Sia structures. (**A**) The mCST-CMP (blue) and mCST-CMP-Sia (orange) structures are superimposed and shown as a top-down view. CMP-Sia is shown as yellow sticks. (**B–E**) The surface of the CMP (shown as yellow sticks) site of the mCST-CMP-Sia structure is shown alone as an orange mesh in (**B**) and (**D**), and superimposed with the CMP site of the mCST-CMP structure, which is shown as a blue surface in (**C**) and (**E**). (**F**) The same comparisons are shown as in (**A**), highlighting the TM1-9 interface. (**G**) Surface views of TMs 1 and 9 are shown for mCST-CMP-Sia (orange) and mCST-CMP (blue). The magenta regions highlight atoms that are within 4 Å of each other at the interface between the two TMs. (**H**) The same comparisons are shown as in (**A**), highlighting the TM5-9-10 bundle.

DOI: https://doi.org/10.7554/eLife.45221.017

However, we did not see strong density for residues 7–14 in the CMP-bound structure and therefore these residues were not modeled (*Figure 2B*).

We propose that these conformational differences may at least partially account for the ~100 fold difference in binding affinity between CMP and CMP-Sia (*Figure 1A*). First of all, the protein distortions needed to accommodate the Sia moiety of CMP-Sia may be energetically costly. In particular, the increased interaction between TMs 1 and 9 and the stabilization of TM1 may lead to a reduction in the conformational entropy of mCST to a degree that is not adequately compensated by enthalpic gains. Similarly, the number of rotatable bonds in CMP-Sia is nearly three times that of CMP (18 versus 7), so stabilization of the Sia moiety of CMP-Sia through interactions with residues on TMs 1 and 4 (*Figures 3C* and *4A*) may also result in a greater entropic penalty to the binding free energy. Finally, the larger and differently-shaped CMP-binding pocket may weaken the interactions between the protein and the CMP moiety of CMP-Sia by negatively impacting the distance and geometry of the numerous non-covalent bonds between the protein and the CMP moiety of CMP-Sia.

An additional component of the differential binding affinity could be related to the reduced charge of the phosphate group in CMP-Sia versus CMP. The highest $pK_a$ of the phosphate group of CMP is 6.3, meaning that it will predominantly have a −2 charge at neutral pH. However, the phosphate group of CMP-Sia only has one titratable oxygen with a very low pKa, so it will only have a −1 charge (although CMP-Sia's overall charge is still −2 on account of the C-1 carboxyl group on the Sia moiety). Therefore, considering the important role that the phosphate group plays in mediating CMP binding to the protein, as described above, a reduction in the phosphate group's charge state

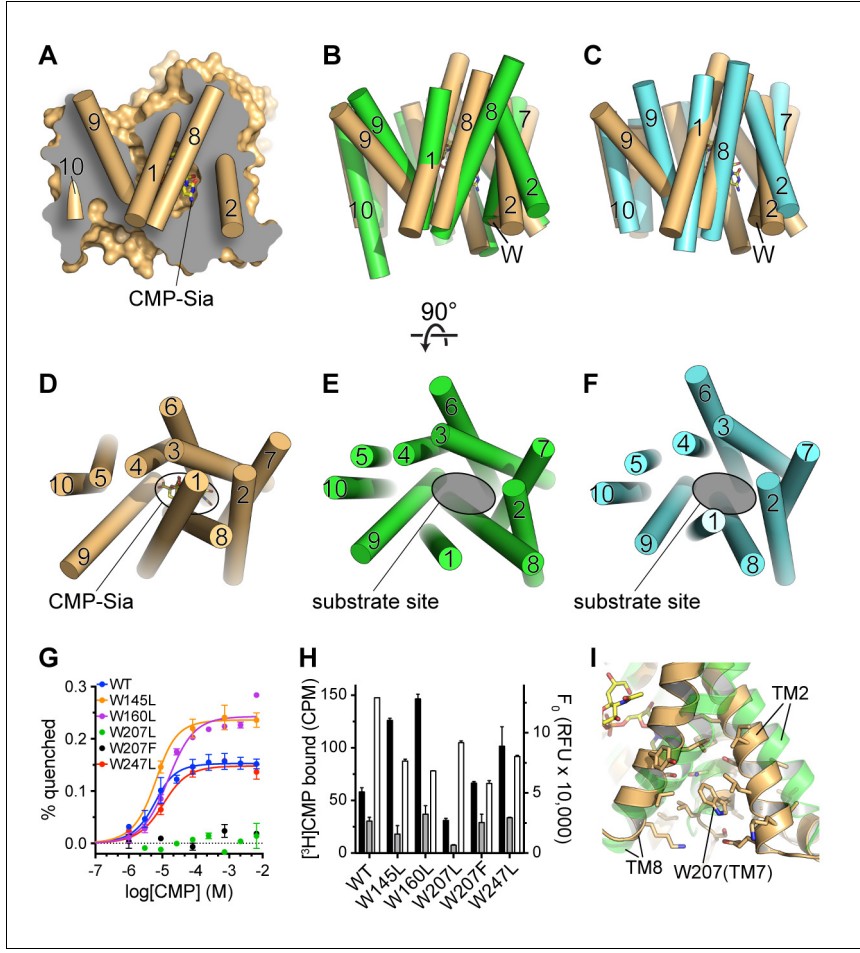

**Figure 5.** Comparison of substrate accessibility for DMT protein structures, and characterization of mCST's intrinsic tryptophan fluorescence. (A) The surface representation of the mCST-CMP-Sia structure from *Figure 2D* is reproduced. The front half of the protein that was removed to make the sliced view is put back on top of the plane of the slice with the alpha-helical TMs represented as cylinders. The TMs are numbered. (B and C) The structures of Vrg4 (B, green), and YddG (C, cyan) are superimposed with the mCST-CMP-Sia structure (orange). These side view orientations are very similar to what is shown in (A) and identical to the view in *Figure 2A and B*. In (A–C), the top and bottom are the lumenal and cytosolic sides, respectively. (D–F) Top-down views from the lumenal side of the Golgi membrane are shown for the mCST-CMP-Sia structure (D, orange), Vrg4 (E, green), and YddG (F, cyan). This orientation is the same as shown in *Figure 2C and F*. The CMP-Sia binding site is highlighted in panel (D) and the equivalent substrate-binding sites are indicated in panels (E) and (F). In panels (B–F), all TMs are also represented as cylinders and TMs are numbered. (G) Quenching of the intrinsic tryptophan fluorescence of wild-type and mutant mCST is shown as a function of CMP concentration. The lack of any quenching for either Trp207 mutant indicates that it is sensing the CMP-induced conformational changes. (H) Left axis: a SPA assay was used to determine the amount of CMP bound to wild-type and mutant mCST to give an indication as to the relative CMP affinity for the tryptophan mutants compared to wild-type mCST. The black bars represent binding in the presence of 30 nM [3H]CMP whereas the gray bars represent binding with an additional 5 µM unlabeled CMP added. CMP affinities, which were all very similar, were estimated by fitting the data to a one-site model and are listed in *Table 1*. These results show that the Trp207 mutants still bind CMP. Right axis: Tryptophan fluorescence in the absence of any ligand is shown for wild-type and mutant mCST. These results show that the initial, ligand-free fluorescence of the Trp207 mutants is not significantly different from wild-type. The symbols and bars in (G) and (H) represent the mean ± SEM for n = 2. (I) The location and local environment of Trp207 of mCST is shown (orange) and compared to the structure of Vrg4 (green). In (B and C) the location of Trp207 is also indicated with the 'W' label.

DOI: https://doi.org/10.7554/eLife.45221.018

may manifest as a lower substrate binding affinity. In support of this, elegant studies on the effect of

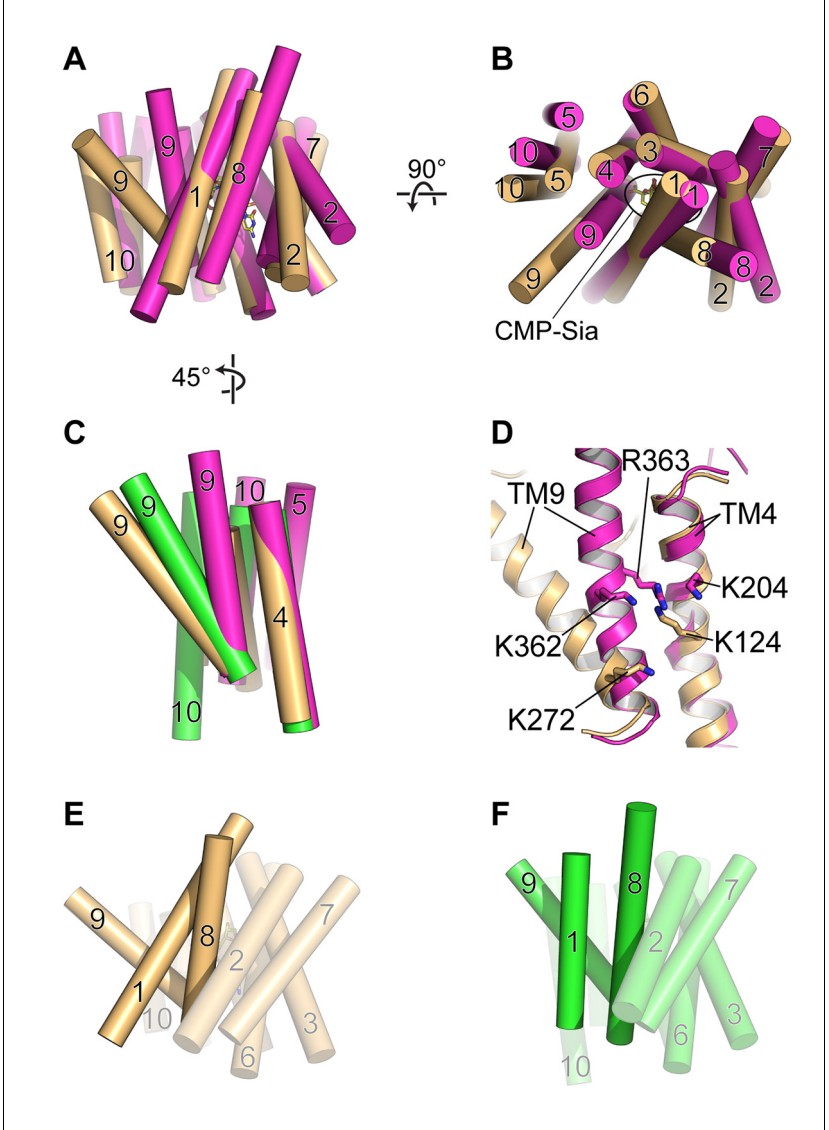

**Figure 6.** Comparison of DMT protein antiport coupling mechanisms. (**A and B**) The mCST-CMP-Sia structure (orange) is superimposed with the GsTPT2 structure (magenta). A side view (**A**) and top-down view from the lumen (**B**) are shown and these orientations are the same as shown in *Figure 5B–5F*. In the side view in (**A**), the top part of the panel is the lumenal/extracellular side for mCST and GsTPT2, respectively. The bottom is the cytoplasmic side. In (**B**) the CMP-Sia binding site is indicated. (**C**) GsTPT2 (magenta) and Vrg4 (green) are superimposed with the mCST-CMP-Sia structure (orange) to show the relative orientation of TM9 with respect to TM4 in these structures. This is a side view as in panel (**A**) but rotated approximately 45° clockwise about a vertical axis. TMs 1–3 and 6–8 are hidden for clarity. (**D**) A close-up view of the same orientation as shown in (**C**) to show the position of Lys and Arg residues on TMs 4 and 9 in the GsTPT2 (magenta) and mCST-CMP-Sia structures (orange). (**E and F**) A similar 45°-rotated side view is shown for the mCST-CMP-Sia structure (**D**, orange) and Vrg4 (**E**, green). All TMs are shown but TMs 1, 8, and 9 are highlighted to show their differential arrangement in the two structures.
DOI: https://doi.org/10.7554/eLife.45221.019

pH on UDP-GlcNAc transporter activity showed that monoanionic UMP is a poor substrate compared to dianionic UMP (*Waldman and Rudnick, 1990*).

## Mechanism of substrate accessibility between mCST and the golgi lumen

As mentioned above, we describe our structures of mCST as partially-occluded because the constriction that separates the substrate-binding site from the Golgi lumen is most likely too small to allow the free passage of a substrate. In order to understand how a substrate can freely exchange between the substrate-binding site and the Golgi lumen, it is first important to understand how the constriction is formed. This is illustrated in *Figure 5A and D* where it can be seen how the constriction is primarily formed by the lumenal end of TM9 being dissociated from the core of the protein. However, the lumenal ends of TMs 1 and 8 are still engaged with the core of the protein to form a lid over the substrate-binding pocket.

Therefore, we propose that the lumenal ends of TMs 1 and 8 must also move away from the core of the protein to allow the substrate-binding cavity to be freely accessible from the lumen. Such a fully-open lumen-facing state may resemble the conformations seen in the structures of other DMT proteins that were determined in the absence of a substrate: the fungal GDP-mannose transporter, Vrg4 (*Parker and Newstead, 2017*), and the bacterial amino acid transporter, YddG (*Tsuchiya et al., 2016*). As shown in *Figure 5E and F*, compared to mCST, both TMs 1 and 8 in Vrg4 and YddG are dissociated from the core of the protein to provide full accessibility to the substrate-binding site.

The concept that mCST may transition between a substrate-free fully-open state and a substrate-bound partially-occluded state is supported by characterization of mCST's intrinsic tryptophan fluorescence (*Figure 5G and H*). We found that when CMP binds mCST, the intrinsic tryptophan fluorescence of mCST is significantly quenched. We further showed that this is due to only one of the four

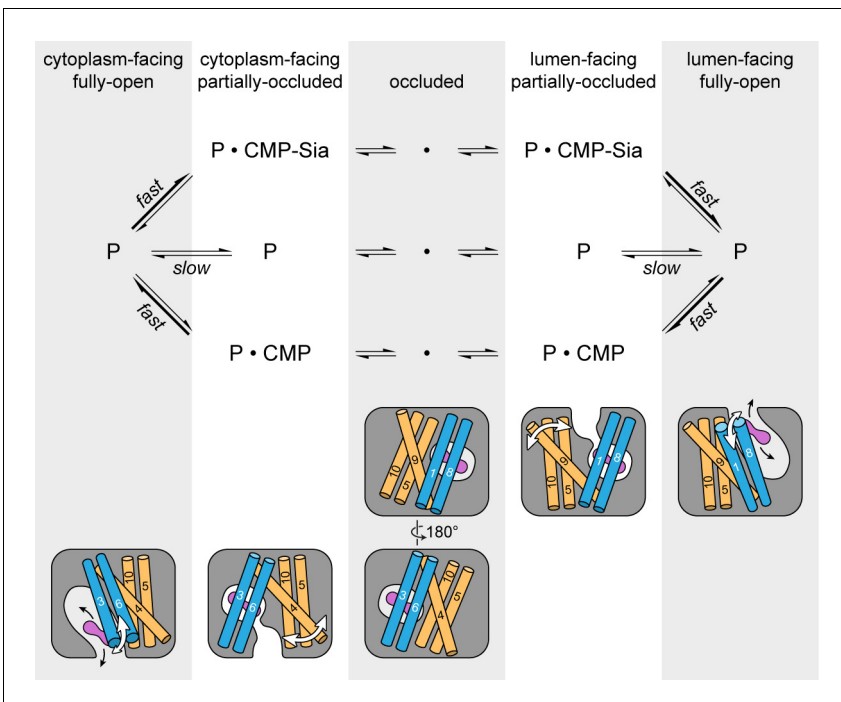

**Figure 7.** Schematic of the proposed transport cycle of mCST. The top row indicates the five predominant conformational states that mCST is expected to occupy. Below this is a simple kinetic model of the mCST transport cycle. 'P' represents the protein and the substrate that is bound in each state is indicated. A cartoon representation for each state is shown at the bottom. The main domains (with numbered TMs) that change conformation to yield each state are highlighted, and the conformational changes that occur in these domains as the protein transitions between different states are indicated by the white arrows. The purple peanut represents a substrate. The alternating gray and white column bars that span the length of the figure are added as visual aids to help associate the state names with the cartoon depictions and their position in the kinetic scheme.
DOI: https://doi.org/10.7554/eLife.45221.020

tryptophans in mCST, Trp207, which is located on the cytoplasmic end of TM7. Since tryptophan fluorescence is sensitive to the polarity of the local environment, these results are consistent with the region around this section of TM7 changing conformation upon substrate binding. Indeed, a comparison of the partially-occluded conformation of mCST with the fully-open conformations of Vrg4 and YddG show that the cytoplasmic ends of TMs 2 and 8, which flank Trp207, would be expected to undergo significant conformational rearrangements as the protein transitions from the fully-open to the partially-occluded state (*Figures 5B, C and I*).

## Structural basis for obligatory antiport

mCST appears to be a nearly-obligatory antiporter – that is, very little CMP-Sia uptake is observed unless vesicles are pre-filled with CMP (*Figure 1B*). This phenomenon, which has been previously described in the literature for CST and several other NSTs, has also been referred to as trans-stimulation or trans-acceleration (*Capasso and Hirschberg, 1984*; *Milla and Hirschberg, 1989*; *Tiralongo et al., 2006*; *Waldman and Rudnick, 1990*). Further structural comparison of mCST with other DMT protein structures provides insight into how the presence of a substrate on the lumenal side of the protein is coupled to the rate of substrate uptake from the cytoplasm.

The algal triose phosphate transporter GsTPT2 is another DMT protein that is also an obligatory antiporter. The structure of GsTPT2 was solved in complex with inorganic phosphate (Pi) and was observed to be in a fully-occluded state (*Lee et al., 2017*). This structure is generally similar to the partially-occluded state of mCST, with the major difference being that the extracellular (lumenal for mCST) end of TM9 is associated with the core of the protein, thus blocking the substrate-binding cavity from having any access to either side of the membrane (*Figure 6A and B*).The Pi molecule in the GsTPT2 structure is coordinated by Lys and Arg residues coming from the middle of TMs 4 and 9 (*Figure 6C and D*). Based on this, the authors propose the following mechanism to describe the strict requirement of obligatory substrate antiport. Formation of the occluded state is driven by binding of the negatively-charged Pi substrate which allows close approximation of the positive charges of the Lys and Arg residues. However, in the absence of a substrate, electrostatic repulsion between these Lys and Arg residues will drive apart TMs 4 and 9 to stabilize a fully-open conformation and prevent the protein from undergoing a conformational transition to face the other side of the membrane.

mCST also has lysines on TMs 4 and 9 – one on each – which are involved in coordinating the phosphate of CMP and CMP-Sia (*Figure 6D*); however, Lys272 is on the cytoplasmic end of TM9. This part of TM9 is not expected to dramatically change conformation between the substrate-bound partially-occluded state and the substrate-free fully-open state, as described in the previous section (*Figures 5A–5F* and *6C*). Furthermore, if mCST were to adopt a fully-occluded conformation that resembles what is seen in GsTPT2, then the cytoplasmic end of TM9 would essentially act as a mostly-stationary hinge during a transition to this state from either the fully-open or partially-occluded states (*Figure 6C and D*). Therefore, we do not think that electrostatic repulsion between lysines on TMs 4 and 9 plays a major role in regulating the stability of different lumen-facing conformational states of mCST.

We instead propose that, in the absence of a substrate, the partially-occluded state of mCST may be less stable than the fully-open state by virtue of the differential arrangement of TMs 1, 8, and 9. TM1 in the partially-occluded state of mCST is mostly orthogonal to TMs 8 and 9, whereas it is more parallel in the fully-open state of Vrg4 (*Figure 6E and F*). This results in significantly less surface area being buried among TMs 1, 8, and 9 in the partially-occluded versus fully-open states (991, 1037, and 1531 Å$^2$ for mCST-CMP, mCST-CMP-Sia, and Vrg4 respectively). The additional ~500 Å$^2$ of buried surface area between TMs 1, 8, and 9 in the fully-open conformation may contribute to the stabilization of this state. This may present enough of a free-energy barrier that would impair the protein from undergoing the necessary structural rearrangements needed to face the cytoplasmic side.

We further propose that substrate binding would lower this barrier by favoring the conversion to the partially-occluded state. This may be driven by a bound substrate interacting with residues on the core of the protein as well as with residues on TMs 1 and 8. For instance, in addition to interactions with residues on TMs 2–4, 6, and 9 as shown in *Figure 3*, CMP also interacts directly, or through structured waters, with residues Ala253, Gly257, Thr260, and Ser261 on TM8 (*Figure 3A* and *Figure 2—figure supplement 1I and J*); and CMP-Sia interacts with these same residues on TM8 as well as with Met20, Ala24, and Tyr27 on TM1 (*Figures 3B, C* and *4A*). The combination of

these protein-substrate interactions along with additional protein-protein interactions between the lumenal ends of TMs 2–4 and the lumenal ends of TMs 1 and 8 (*Figures 2C* and *5D*) may favor the formation of the partially-occluded state by sufficiently offsetting the otherwise unfavorable loss of ~500 Å$^2$ of buried surface area among TMs 1, 8, and 9. The formation of a partially-occluded state may be a necessary intermediate state that allows the protein to transition to face the cytoplasmic side.

For CST, it has been reported that lumenal CMP is 2–3 fold more effective at trans-stimulation of CMP-Sia uptake than an equivalent lumenal concentration of either UMP or CMP-Sia (*Chiaramonte et al., 2001*; *Tiralongo et al., 2006*). One possible reason for this may be the large difference in binding affinities between CMP and either UMP or CMP-Sia (*Figure 1A*). In other words, if we assume that we are measuring binding affinities towards the lumen-facing partially-occluded state, then for a given sub-saturating concentration of a substrate in the lumen, CMP will have a greater fractional occupancy of the lumen-facing partially-occluded state.

## Alternating access transport model

As mentioned previously, mCST has symmetry-related structural repeats (*Figure 2—figure supplement 3*) – similar to what is seen in other DMT proteins (*Lee et al., 2017*; *Parker and Newstead, 2017*; *Tsuchiya et al., 2016*) as well as in many other transporter families (*Drew and Boudker, 2016*; *Forrest, 2013*). Although there are exceptions, one commonly-observed alternating access mechanism that has been observed in these types of transporters is that symmetry-related conformational changes are involved in alternately exposing the protein to either side of the membrane (*Drew and Boudker, 2016*; *Forrest, 2013*). Symmetry-related exchange mechanisms have been proposed for other DMT proteins (*Lee et al., 2017*; *Parker and Newstead, 2017*; *Tsuchiya et al., 2016*) and we think that mCST may operate in a similar fashion (*Figure 7*).

As the protein transitions from a lumen-facing partially-occluded state to face the cytoplasmic side, it may first pass through a fully-occluded state that resembles the GsTPT2 structure. As shown in the structural comparison between mCST and GsTPT2, this would not only involve the lumenal-end of TM9 associating with the core of the protein but also a rearrangement of TMs 5 and 10 (*Figure 6A and B*). As discussed when comparing the structural differences between the CMP and CMP-Sia bound mCST structures, TMs 5, 9, and 10 do have a propensity to move independently of the rest of the protein (*Figure 4H*). The role of these TMs in substrate transport is supported by the fact that mutations of residues at the interfaces of these TMs (in CST and other human NST isoforms) form the basis for several diseases (*Figure 3—figure supplement 1* and *Table 3*). As the protein continues to transition, TM4 may move away from the protein core to form a cytoplasmic-facing partially-occluded state. Formation of a cytoplasmic-facing fully-open state would then involve the cytoplasmic ends of TMs 3 and 6 moving away from the protein core (*Figure 7*). After substrate exchange, it's possible that similar conformational transitions would happen in the reverse order to ferry a nucleotide sugar to the Golgi lumen.

Cytoplasmic concentrations of CMP and CMP-Sia have been estimated to be between 5–44 μM for CMP and 75–450 μM for CMP-Sia (*Chiaramonte et al., 2001*; *Briles et al., 1977*; *Hauschka, 1973*; *Nakajima et al., 2010*; *Traut, 1994*). On the other hand, lumenal concentrations of NMPs are much higher. CMP concentrations in particular have been estimated to be in the range of 0.167–2.1 mM (*Fleischer, 1981*; *Waldman and Rudnick, 1990*). This concentration gradient of NMPs across the Golgi membrane accounts for the observation that CST, like many other NSTs, is a secondary active transporter and can concentrate nucleotide sugars inside the Golgi lumen (*Hirschberg et al., 1998*). This is an important property of this system as many glycosyltransferases have $K_m$'s for nucleotide sugars that can approach mM concentrations and sialyltransferase $K_m$'s in particular range from 0.05 to 3.2 mM (*Gupta et al., 2016*). The kinetics of substrate efflux from the Golgi through CST are not well defined; however, high lumenal CMP concentrations coupled with CMP's relatively high binding affinity for the partially-occluded state will ensure efficient conversion of CST back to a cytoplasmic-facing state. Although reported values for the $K_i$ of CMP inhibition of CMP-Sia uptake (*Chiaramonte et al., 2001*; *Tiralongo et al., 2000*) are similar to reported $K_m$ values of CMP-Sia uptake (*Aoki et al., 2003*; *Chiaramonte et al., 2001*; *Milla and Hirschberg, 1989*; *Tiralongo et al., 2006*), the approximately 10-fold higher cytoplasmic concentrations of CMP-Sia will ensure that CMP-Sia is transported into the Golgi lumen at a high fraction of $V_{max}$.

## Conclusion

These structures of mCST represent the first high-resolution structures of an NST in complex with its physiological substrates. Analysis of the structures and homology models of related NSTs explains the elegant mechanisms through which NSTs discriminate between different nucleotides and nucleotide-sugar conjugates. Comparing mCST structures to other DMT protein structures provides insight into the overall transport mechanism and explains the mechanisms of obligatory antiport and trans-stimulation, which is further supported by characterization of the differential binding affinities of CMP and CMP-Sia as well as by identifying the tryptophan that senses CMP-induced conformational changes in mCST. These analyses explain how disease-causing mutations in human NST genes affect transport activity and provide a framework for pharmacologically targeting NSTs.

# Materials and methods

**Key resources table**

| Reagent type (species) or resource | Designation | Source or reference | Identifiers | Additional information |
|---|---|---|---|---|
| Gene (Mus musculus) | *Slc35a1*;CMP-sialic acid transporter; mCST;mCST∆C | Biobasic | Uniprot: Q61420 | |
| Recombinant DNA reagent | pPICZ | ThermoFisher | cat# V19020 | |
| Strain, strain background (Pichia pastoris) | SMD1168H | ThermoFisher | cat# C18400 | |
| Chemical compound, drug | Cytidine 5'-monophosphate; CMP | Sigma-Aldrich | cat# C1006 | |
| Chemical compound, drug | Cytidine 5'-monophospho-N-acetylneuraminic acid; CMP-sialic acid; CMP-Sia | Carbosynth | cat# MC04391 | |
| Chemical compound, drug | [3H] CMP | American Radiolabeled Chemicals | cat# ART0342 | |
| Chemical compound, drug | [3H] CMP-sialic acid | American Radiolabeled Chemicals | cat# ART0147 | |
| Chemical compound, drug | cytidine | Sigma-Aldrich | cat# C122106 | |
| Chemical compound, drug | sialic acid | Fisher | cat # ICN15142450 | |
| Chemical compound, drug | Uridine 5'-monophosphate; UMP | Sigma-Aldrich | cat# U6375 | |
| Chemical compound, drug | Guanosine 5'-monophosphate; GMP | Sigma-Aldrich | cat# G8377 | |
| Peptide, recombinant protein | Antarctic phosphatase | New England Biolabs | cat# M0289S | |
| Software, algorithm | XDS | doi: 10.1107/S0907444909047337 | SCR:015652 | |
| Software, algorithm | Pointless | doi: 10.1107/S0907444491003982 | SCR:014218 | |

*Continued on next page*

*Continued*

| Reagent type (species) or resource | Designation | Source or reference | Identifiers | Additional information |
|---|---|---|---|---|
| Software, algorithm | Aimless | doi: 10.1107/S0907444913000061 | SCR:015747 | |
| Software, algorithm | CCP4i | doi: 10.1107/S0907444910045749 | SCR:007255 | |
| Software, algorithm | XSCALE | doi: 10.1107/S0907444909047337 | SCR:015652 | |
| Software, algorithm | UCLA anisotropy server | doi: 10.1073/pnas.0602606103 | | https://services.mbi.ucla.edu/anisoscale/ |
| Software, algorithm | HKL2Map | doi: 10.1107/S0021889804018047 | | |
| Software, algorithm | SHELX | doi: 10.1107/S0108767307043930 | SCR:014220 | |
| Software, algorithm | SHARP | doi: 10.1107/S0907444903017694 | | |
| Software, algorithm | DM | doi: 10.1107/S090744499500761X | | |
| Software, algorithm | Coot | doi: 10.1107/S0907444910007493 | SCR:014222 | |
| Software, algorithm | Refmac | doi: 10.1107/S0907444911001314 | SCR:014225 | |
| Software, algorithm | Phaser | doi: 10.1107/S0021889807021206 | SCR:014219 | |
| Software, algorithm | MolProbity | doi: 10.1107/S0907444909042073 | SCR:014226 | |
| Software, algorithm | Phenix | doi: 10.1107/S0907444909052925 | SCR:014224 | |
| Software, algorithm | PyMOL | pymol.org | SCR:000305 | |
| Software, algorithm | SWISS-MODEL webserver | doi: 10.1093/nar/gky427 | SCR:013032 | https://swissmodel.expasy.org/ |
| Software, algorithm | AutoDock4 | doi: 10.1002/jcc.21256 | SCR:012746 | |

## Molecular biology

Full-length mCST cDNA was synthesized (Bio Basic) and subcloned into the *Pichia pastoris* expression vector pPICZ with a C-terminal PreScission protease site, followed by green fluorescent protein (GFP), and then a His10 tag. The mCSTΔC construct was generated by modifying the full-length construct using site-directed mutagenesis to remove the last 15 residues (322-336).

## Protein expression and purification

Full-length and mutant mCST constructs were expressed in *P. pastoris* and purified as previously described (*Whorton and MacKinnon, 2011*; *Whorton and MacKinnon, 2013*) with a few modifications. Milled cells were solubilized for 1.5 hr at 4°C in the following buffer: 50 mM HEPES pH 7.5, 150 mM NaCl, 0.01 mg/ml deoxyribonuclease I, 0.7 µg/ml pepstatin, 1 µg/ml leupeptin, 1 µg/ml aprotinin, 1 mM benzamidine, 0.5 mM phenylmethylsulfonyl fluoride, and 2% (w/v) n-dodecyl-β-D-maltopyranoside (DDM) (Anatrace, solgrade). The lysate was centrifuged at 35,000 g for 35 min at 4°C to pellet the unsolubilized material. The clarified supernatant was pooled and the pH was adjusted to 7.2 with 5 M NaOH, then added to Talon resin (0.175 ml/g of cells; Clontech) pre-equilibrated in Buffer A (50 mM HEPES pH 7.5, 150 mM NaCl, 0.1% (w/v) DDM (solgrade)) and incubated at 4°C for 2 hr under gentle rotation. 5 mM imidazole was added during binding to the Talon resin.

The resin was washed in batch with five column volumes (cv) of Buffer A with 5 mM imidazole by pelleting at 1250 g for 5 min and re-suspending in fresh buffer. Washed resin was loaded onto a column and further washed with five cv Buffer A + 20 mM imidazole, then two cv Buffer A + 40 mM imidazole at about 1 ml/min using a peristaltic pump. The column was then eluted with Buffer A + 300 mM imidazole. Peak fractions were pooled and 1 mM DTT, 1 mM EDTA was added. PreScission protease was added at 1 µg protease per 20 µg of protein to cut the C-terminal GFP tag overnight at 4°C. The cleaved protein was concentrated in a 50 K MWCO concentrator (Millipore) to run on a Superdex 200 gel filtration column (GE Healthcare) in Buffer B (25 mM HEPES pH 7.5, 150 mM NaCl, 0.1% (w/v) DDM (anagrade), 5 mM DTT, and 1 mM EDTA).

## Oligomeric state determination

We characterized the oligomeric state of purified human CST (hCST) as it was initially one of our most promising constructs but did not grow well-diffracting crystals. Compared to mCST, it has an identical migration time on a gel filtration column (data not shown); therefore, we expect that the results from the following experiments will also be applicable to mCST. The molecular weight of the protein component of the purified hCST-lipid-detergent complex, which has a theoretical molecular weight of 37 kDa, was determined with the 'SEC-UV/LS/RI' approach using the 'three detector' method (Arakawa et al., 1992; Folta-Stogniew, 2006; Hayashi et al., 1989). 250 µl of purified hCST at 0.8 mg/ml was loaded onto a Superdex 200 column equilibrated in Buffer B. The output from the column was then passed through UV absorbance, light scattering, and refractive index detectors. The chromatography and analysis of the data were performed at the Biophysics Resource of Keck Facility at Yale University.

For the cross-linking reactions, either disuccinimidyl suberate (DSS; Thermo Fisher Scientific) or bis(sulfosuccinimidyl)suberate ($BS^3$; Thermo Fisher Scientific) were added at the indicated concentrations to 1 mg/ml purified hCST in Buffer A. The reactions were incubated for 30 min at room temperature (RT) and then quenched by adding 1 M Tris-HCl pH 7.5 to a final concentration of 100 mM. Samples were then incubated for 15 min at RT before being run on a 12% SDS-PAGE gel.

## HPLC analysis and phosphatase treatment of CMP-Sia

CMP-Sia (Carbosynth) purity was assessed by reverse phase HPLC using an established protocol (Nakajima et al., 2010) with some modifications. Briefly, an XSelect CSH C18 column (3.5 µm, 2.1 × 150 mm; Waters) was first equilibrated for at least 30 min (at 0.2 ml/min) in 70% Buffer C (0.1 M $KPO_4$ pH 6.5, 8 mM tetrabutylammonium hydrogensulfate (TBHS)) and 30% Buffer D (100% acetonitrile). A sample method involved first running 100% Buffer C for 10.4 min, then a 0–30% Buffer D linear gradient for 1.6 min, and then maintaining this mobile phase for 15 min before running a linear gradient back to 100% Buffer C over 1 min, and then maintaining this mobile phase for 6 min, all at 0.2 ml/min. For a series of experiments, we would first pre-condition the column by twice injecting 10 µl of Buffer C. We would then inject 10 µl of each CMP-Sia sample. The flow-through was passed through an absorbance detector set to a wavelength of 270 nm to maximize the signal for cytidine-containing compounds. Sialic acid eluted in the void volume and was not detected under these conditions. All samples were kept at 4°C until immediately before injection onto the column. All buffers and chromatographic steps were at RT.

Antarctic phosphatase (AnP, New England Biolabs) was added to CMP-Sia samples to convert all free CMP to cytidine. This typically involved first concentrating the AnP while also reducing the glycerol concentration by diluting to 100 U/ml in AnP buffer (50 mM Bis-Tris-Propane-HCl, pH 6.0, 0.1 mM $ZnCl_2$ and 1 mM $MgCl_2$; New England Biolabs) and then concentrating to 10,000 U/ml with a 10 K MWCO concentrator (Millipore). For co-crystallization experiments, CMP-Sia powder was dissolved with a 6000 U/ml AnP solution to a final concentration of 413 mM. Additional $ZnCl_2$ and $MgCl_2$ were added to a final concentration of 0.2 mM and 2 mM, respectively, and this mixture was then incubated for 28 hr at RT. After confirming the absence of CMP using the HPLC assay (described above), this stock was then used immediately for crystallization experiments. For binding and transport assays, CMP-Sia powder was dissolved into a 10,000 U/ml AnP solution to a final concentration of 100 mM. Additional $ZnCl_2$ and $MgCl_2$ were added to a final concentration of 0.2 mM and 2 mM, respectively, and this mixture was then incubated for 8 hr at RT. The AnP was then removed by filtering this mixture through a 3 K MWCO concentrator (Millipore), which retained the

70 kDa AnP. We determined that this step removed any detectable AnP activity from our stocks (discussed below). The elimination of CMP from CMP-Sia stocks was confirmed by HPLC analysis (described above) before aliquots were frozen and stored at −80˚C.

The rate of CMP production by CMP-Sia hydrolysis in an AnP-treated CMP-Sia sample, after the AnP was removed, was determined in conditions that mimic our binding and transport assay conditions. This was done by incubating 10 mM AnP-treated CMP-Sia in either a buffer at pH 6.5 (0.1 M KPO$_4$ pH 6.5) or a buffer at pH 7.5 (20 mM HEPES pH 7.5 and 0.15 M NaCl) for various times at RT, as shown in *Figure 1—figure supplement 3D–F*. These samples were then kept on ice before diluting to 2 mM in Buffer C and then running on a C18 HPLC column (at RT) as described above. The fraction of each species present in a sample was determined by dividing the area of each peak (at the following elution times: ~3 min for CMP; ~3.8 min for cytidine; and ~5 min for CMP-Sia) by the total combined area of the CMP, cytidine, and CMP-Sia peaks. This method of species quantitation was used to account for small differences in injection volumes between runs. The slope of a linear regression fit of the fraction of each species as a function of time gave the rate of species production/degradation for a given pH. The fraction of CMP present at 0 min is due to a combination of CMP not removed during AnP treatment as well as CMP that is produced during the ~5 min that the sample spends at RT in the pH 6.5 mobile phase during the HPLC run before eluting. These two sources were differentiated by first calculating the amount of CMP produced during the HPLC run using the rate of CMP produced at pH 6.5 as determined above. The residual amount of CMP was taken to have originated from incomplete removal during AnP treatment.

The results from this analysis are shown in *Figure 1—figure supplement 3G–H*. The starting CMP-Sia fraction of 88.68% was used to determine the actual concentration of CMP-Sia in any given sample of AnP-treated CMP-Sia. The relatively slow rate of CMP-Sia degradation (−0.0049 %/min) means that the concentration would only change by 0.15% in 30 min at RT, which was not considered significant enough for analyzing results from transport and binding experiments. On the other hand, while the rate that CMP is produced from CMP-Sia hydrolysis is nearly as slow (0.0051 %/min), this still results in a non-negligible amount of CMP. For transport assays, the relatively low concentrations and short incubation times limit the total CMP concentration to 48 nM at the end of the time course shown in *Figure 1B* and only 26 nM at the highest concentration of CMP-Sia used in the titration shown in *Figure 1C*. As these concentrations are at least 100-fold lower than the reported $K_i$ for CMP inhibition of CMP-Sia transport (*Chiaramonte et al., 2001*), these concentrations of CMP were not considered significant. However, the higher concentrations and long counting times for the SPA assay resulted in higher concentrations of CMP contamination, ranging from 22 nM to 65 µM. We discuss how this was taken into account in the SPA assay section below. Finally, the rate of cytidine production was determined to be essentially 0 %/min, considering the error on the measurement, which demonstrates that the filtration step to remove AnP, described above, is sufficient to eliminate all AnP activity.

## Crystallization

For hanging-drop vapor-diffusion crystallization (HDVD) experiments, purified mCST was first concentrated to 5 mg/ml using a 50 K MWCO concentrator (Millipore) and then CMP (Sigma) was added to 20 mM. This sample was then mixed 1:1 with 200 nl of the crystallization solution (25.6–26.8% PEG 400, 0.1 M Tris pH 8.5, and 0.1 M magnesium acetate) and incubated at 20˚C. Rectangular rod-shaped crystals typically appeared after about five days and would continue to grow for approximately eight more days. The crystals were cryoprotected by increasing the concentration of PEG400 in the well up to 32–35% and incubating for about 16–24 hr. The crystals were then harvested directly from the drop and flash-frozen in liquid N$_2$.

For lipidic cubic phase (LCP) crystallization experiments, mCSTΔC was first purified in Buffer B with 0.03% DDM, then concentrated to 20 mg/ml, and then either CMP was added to 20 mM, or AnP-treated CMP-Sia was added to 18.4 mM (additional AnP was also added to bring the final concentration to 390 U/ml). Either of these samples was then mixed 2:3 with monoolein (Nu-Chek Prep) and then 70 nl of this material was deposited on a glass slide (Molecular Dimensions). 600 nl of the crystallization solution (26.7–30% PEG 300, 0.1 M MES pH 6.5, and 0.1 M NaCl for CMP; 28.05% PEG 350 MME, 40 mM Tris pH 8.0, and 40 mM NaCl for CMP-Sia) with either 10 mM CMP or 8 mM CMP-Sia (additional AnP was also added to bring the final concentration to 173 U/ml in 600 nl) was then added. The drop was sealed with a glass slide on the top and incubated at 20˚C. Long needle-

shaped crystals typically appeared after 1–2 days and would grow for another 2–4 days. The crystals were then harvested directly from the drop and flash-frozen in liquid N$_2$.

## Structure determination

All diffraction data were processed with XDS (*Kabsch, 2010*) and further analyzed using Pointless (*Evans, 2011*) and Aimless (*Evans and Murshudov, 2013*). CCP4i (*Winn et al., 2011*) was used for project and job organization. Diffraction data for the HDVD mCST-CMP native crystals were collected at the Advanced Photon Source (APS) beamline 23ID-B and the Advanced Light Source (ALS) beamline 5.0.2, using X-ray wavelengths of 1.03313 Å and 1.0 Å, respectively. Data from five different crystals were merged using XSCALE. The diffraction data were highly anisotropic, so they were truncated and scaled using the UCLA anisotropy server (*Strong et al., 2006*). Data for Hg and Pt derivatives were collected at the APS 21-D and ALS 5.0.2 beamlines, respectively, using 1.0052 Å and 1.07234 Å wavelengths, respectively. These derivatives were prepared by adding ~0.5 µl of 10 mM solutions of either CH$_3$HgCl or (NH$_4$)$_2$PtCl$_4$, prepared in a buffer that mimicked the mother liquor, to crystal drops and soaking for either 10 min or 24 hr, respectively before harvesting. Hg or Pt sites were located with HKL2Map (*Pape and Schneider, 2004*) and the SHELX (*Sheldrick, 2008*) suite of programs. Initial phases were determined using the MIRAS method in SHARP (*Bricogne et al., 2003*). Phases were extended and improved using solvent flattening, histogram matching, and 2-fold non-crystallographic symmetry averaging using the program DM (*Cowtan, 1994*). An atomic model was built using Coot (*Emsley et al., 2010*) and improved through iterative cycles of refinement using Refmac (*Murshudov et al., 2011*). The register for nearly all of the TMs could be determined either from the presence of a labeled cysteine or bulky side chains.

Diffraction data for the LCP crystals were collected at the APS 23ID-B and 23ID-D beamlines using a wavelength of 1.0332 Å. Data from 26 different crystals were merged using XSCALE. The LCP mCST-CMP crystal structure was solved by molecular replacement using Phaser (*McCoy et al., 2007*), using the HDVD mCST-CMP crystal structure model as a search model. Iterative cycles of model building in Coot and refinement using Refmac were used to add more than 40% additional atoms to the model, including CMP. The model includes residues 15–317, but lacks residues 164–167 of the loop between TMs 5 and 6. This model was also then refined against the HDVD mCST-CMP dataset, which yielded the model shown in *Figure 2—figure supplement 2A–C*. The statistics for this model are shown in *Table 2*. The LCP mCST-CMP-Sia crystal structure was also solved by molecular replacement using Phaser, using the LCP mCST-CMP crystal structure model as a search model. Iterative cycles of model building in Coot and refinement using Refmac were used to build the more ordered N-terminus and to add CMP-Sia. The model includes residues 7–317, but lacks residues 161–167 of the loop between TMs 5 and 6. All models were validated using MolProbity (*Chen et al., 2010*) as implemented in Phenix (*Adams et al., 2010*). The HDVD mCST-CMP, LCP mCST-CMP, and LCP mCST-CMP-Sia models had 98.3, 99.3, and 98.0%, respectively, of their residues in the preferred region of a Ramachandran plot and no outliers. Figures were prepared using PyMOL (*Schrodinger, 2015*).

## Transport assay

The purification protocol for mCST constructs used for transport assays was altered slightly to achieve higher purity. This involved dialyzing the PreScission protease digestion overnight at 4°C against Buffer A to remove the imidazole. The cleaved protein was then run over a 1 ml Talon column equilibrated in Buffer A. The flow through as well as two subsequent one cv washes were then concentrated and run on a Superdex 200 column as described above. Purified constructs were then incorporated into lipid vesicles comprised of yeast polar lipid (YPL) extract (Avanti Polar Lipids). A typical reconstitution entailed first drying down 40 mg of YPL from a chloroform suspension using a stream of argon, followed by two washes with pentane and then placing the dried film of lipids in a vacuum desiccator overnight. The lipid film was re-suspended at 11.1 mg/ml in Buffer E (20 mM HEPES pH 7.5 and 0.1 M KCl) and then sonicated to form small unilamellar vesicles. DDM was then added to a final concentration of 5 mM and incubated for 1 hr at RT. 200 µg purified mCST (or an equivalent volume of the gel filtration buffer for protein-free vesicles), either with or without 300 µM CMP (final concentration), was then added. Additional gel filtration buffer was added to bring the final lipid concentration to 10 mg/ml and the mixture was then incubated for 1 hr at 4°C.

Proteoliposomes were formed by removing DDM with three successive additions of Bio-Beads (SM-2; Bio-Rad) at 100 mg/ml. For the first two additions, the Bio-Beads were incubated for 2 hr at 4°C under gentle rotation followed by a third incubation overnight. The liposomes were then aliquoted, flash frozen in liquid $N_2$, and stored at −80°C. We were typically able to incorporate about 60% of the starting amount of protein into the vesicles, which we determined by solubilizing small amount of vesicles with DDM and then running on a gel filtration column (data not shown).

For transport assays, vesicles were thawed and extruded 10 times through a 0.4 µm Whatman Nuclepore filter (GE Healthcare). For a typical reaction, 45 µl of extruded vesicles was used. External CMP was removed by pelleting the vesicles by ultracentrifugation (194,800 g for 60 min), removing the supernatant, washing the pellet with Buffer E, and then resuspending the vesicles in 20 µl of cold Buffer E and kept on ice. Transport was initiated by bringing the volume of the vesicles to 50 µl using 30 µl of RT Buffer E containing the indicated concentration of CMP-Sia which contained 30–50 nM [$^3$H]CMP-Sia (20 Ci/mmol; American Radiolabeled Chemicals). The mixture was then incubated at RT for the indicated time for time-course experiments or for 30 s to determine the initial velocity for substrate titration experiments. Transport was then stopped by adding 0.6 ml ice-cold Buffer E and storing on ice. The transport reactions were then filtered through 0.22 µm mixed cellulose ester membranes (Millipore) and washed with 3 × 2 ml ice-cold Buffer E. Counts from protein vesicles at 0 min and 4°C (similar to background counts from reactions with protein-free vesicles) were subtracted from the total counts from the protein-containing vesicles to determine specific counts. Transport data were fit to a Michaelis-Menten model to determine $K_m$ and $V_{max}$.

## Scintillation proximity binding assay

For scintillation proximity assays (SPA), mCST constructs were purified as described above, except the GFP-His10 tag was not removed and DTT and EDTA were eliminated from Buffer B during the final size exclusion chromatography step. A typical binding experiment involved combining 2 µM purified protein, 0.4 mg Copper HIS-Tag PVT SPA beads (PerkinElmer), 30 nM [$^3$H]CMP (20 Ci/mmol; American Radiolabeled Chemicals) and various concentrations of cold substrates in a final volume of 50 µl. Non-specific counts were determined by setting up identical reactions without protein. The protein was added to beads for 30 min on ice to allow the protein to bind the beads before the ligands were added. The reactions were prepared in 96-well, white clear-bottom Isoplates (PerkinElmer) and incubated for 5 min on ice followed by a minimum incubation of approximately 30 min at RT before being counted in a Wallac MircoBeta TriLux scintillation counter (PerkinElmer) using the 'SPA-counting mode.' For the CMP titration, the relatively high affinity of CMP meant that we had to titrate CMP close to the concentration of mCST used in the assay, so the following homologous binding model that takes into account ligand depletion was used to determine the fraction of CMP-bound mCST ($f_P$):

$$f_P = \frac{S + S' + K + P - \sqrt{(S + S' + K + P)^2 - 4SP}}{2P} \quad (1)$$

where $S$ is the concentration of cold CMP that was titrated, $S'$ is the concentration of [$^3$H]CMP, $K$ is the $K_d$ for CMP, and $P$ is the concentration of mCST. $K$ was determined by fitting the experimental data to this model. Ligand depletion only had a minor effect on the interpretation of our results as $K_d$'s that were determined with this model only differed by approximately 1.3-fold from $IC_{50}$'s determined from a simple one-site binding model. For other titrations, the fraction of mCST bound to the indicated substrate ($f_P$) was modeled using the following equation:

$$f_P = \frac{S1}{S1 + K_{S1}\left(1 + \frac{S2}{K_{S2}}\right)} \quad (2)$$

where $S1$ and $S2$ are the concentrations of substrate 1 (the titrated substrate) and substrate 2 (either just [$^3$H]CMP or the combination of [$^3$H]CMP and cold CMP generated from CMP-Sia hydrolysis, which is discussed below), respectively. $K_{S1}$ and $K_{S2}$ are the $K_d$'s of substrate 1 and 2, respectively. $K_{S2}$ is the $K_d$ for CMP, which was determined using *Equation 1* as described above. $K_{S1}$ was determined by fitting the experimental data to this model. For CMP-Sia titrations, the final concentration of contaminating CMP at each assay point was determined by multiplying the rate of CMP

production, as determined above, by the total time that elapsed between when the assay plate was placed at RT to when an individual well was counted. This elapsed time typically varied from between 30–60 min, as each well was counted for 1 min. While the concentration of CMP in some wells was relatively high, the $K_d$'s that we determined through this method only differed by approximately 2-fold from $IC_{50}$'s determined from a simple one-site binding model where the generation of CMP from CMP-Sia hydrolysis is ignored.

## Tryptophan fluorescence assay

The effect of CMP on the intrinsic tryptophan fluorescence of wild-type and mutant mCST was measured using a Photon Technology International dual-monochromator fluorometer. Since cytidine's absorption spectrum ($\lambda_{max}$=270) partially overlaps that of tryptophan's ($\lambda_{max}$=280), this leads to an apparent fluorescence quenching termed the inner-filter effect (*Lakowicz, 2006*). To account for this, we first minimized this effect by using a slightly higher excitation wavelength (300 nm) that is not as well absorbed by cytosine. What little inner-filter effect remained could be measured and corrected for by performing the CMP titration in the presence of mCST denatured with sodium dodecyl sulfate (SDS). A typical experiment involved putting 400 µl of 6 µM mCST into a quartz cuvette and then recording the emission spectrum from 315 to 500 nm. Subsequent spectra would then be recorded after adding 0.4–2.5 µl of a concentrated stock of CMP to achieve the desired concentration. An identical titration was performed in the presence of 2.5% SDS. A spectrum collected from a buffer-only sample was used for subtracting the background fluorescence from the spectra of protein samples. The $\lambda_{max}$ for the tryptophan fluorescence of mCST was around 328 nm and did not change as a function of CMP concentration. Peak height was determined by averaging the values from 321 to 335 nm. To correct for the inner-filter effect, a correction factor for each CMP concentration was first calculated by dividing the peak height of the SDS-treated sample without CMP by the peak heights of each SDS-treated sample with various concentrations of CMP. The peak heights for the native protein samples were then multiplied by this correction factor. Correction values were typically very small up to 700 µM CMP (ranging between 1–1.03) and increased slightly to ~1.2 for 7 mM CMP. The corrected peak heights were then used to calculate fractional quenching as a function of CMP concentration. The quenching data were fit to a version of *Equation 1* that lacked the *S'* terms to determine the $K_d$'s for CMP.

## Homology modeling and substrate docking

AZTMP was manually docked into the mCST-CMP structure. Structural homology models of UGT and NGT were generated using the SWISS-MODEL web server (*Waterhouse et al., 2018*), with the mCST-CMP-Sia structure used as a template. UDP-sugars were docked into these models using AutoDock4 (*Morris et al., 2009*) with the flexible side chain covalent docking method, as previously described (*Bianco et al., 2016*). To do this, UMP was first placed in the same pose that CMP is found in the mCST-CMP-Sia structure and it was set to act as part of the rigid component of the protein. This greatly reduced the computation required and is justified by the nearly-complete conservation of the residues that comprise the nucleotide binding site in mCST, UGT, and NGT. The sugar moiety of the UDP-sugar was then treated as a flexible residue. The residues that line the sugar binding site were also set to be flexible while the rest of the protein was kept rigid. The ligand poses shown in *Figure 3—figure supplement 2D–F* are the lowest-energy poses out of 1000 docking runs.

## Data availability

Atomic coordinates and structure factors are available in the Protein Data Bank (PDB) with entries 6OH4 (HDVD mCST-CMP), 6OH2 (LCP mCST-CMP), and 6OH3 (LCP mCST-CMP-Sia).

## Acknowledgements

We thank the following beamline staff for assistance with remote data collection: M. Becker, N. Sanishvili, N. Venugopalan (GM/CA, APS); S. Anderson (LS-CAT, APS); M. Allaire and N. Smith (5.0.2, ALS). We thank E. Folta-Stogniew at the Biophysics Resource of Keck Facility at Yale University for performing the SEC-LS/UV/RI experiment, which was supported by NIH Award Number 1S10RR023748-0. We also thank D. Farrens for helpful discussions and use of a fluorometer; G.

Mandel for use of a fluorescence plate reader; E. Gouaux for helpful discussions and use of crystallization robotics and scintillation counters; K. Hartfield, S.-L. Shyng, and F. Valiyaveetil for comments on the manuscript; K. Beeson and R. Posert for help with construct screening and cell culture, and L. Vaskalis for help with figure preparation. This work was supported by NIH grant R01GM130909.

## Additional information

### Funding

| Funder | Grant reference number | Author |
|---|---|---|
| Oregon Health and Science University | | Shivani Ahuja Matthew R Whorton |
| National Institutes of Health | R01GM130909 | Shivani Ahuja |

The funders had no role in study design, data collection and interpretation, or the decision to submit the work for publication.

### Author contributions

Shivani Ahuja, Data curation, Formal analysis, Validation, Investigation, Methodology, Writing—review and editing; Matthew R Whorton, Conceptualization, Data curation, Formal analysis, Supervision, Funding acquisition, Validation, Investigation, Methodology, Writing—original draft, Project administration, Writing—review and editing

### Author ORCIDs

Shivani Ahuja (ID) http://orcid.org/0000-0002-0123-711X
Matthew R Whorton (ID) http://orcid.org/0000-0002-9915-7467

### Decision letter and Author response

Decision letter https://doi.org/10.7554/eLife.45221.029
Author response https://doi.org/10.7554/eLife.45221.030

## Additional files

### Supplementary files

• Transparent reporting form
DOI: https://doi.org/10.7554/eLife.45221.021

### Data availability

Atomic coordinates and structure factors have been deposited in the Protein Data Bank (PDB) with entries 6OH2, 6OH3, and 6OH4.

The following datasets were generated:

| Author(s) | Year | Dataset title | Dataset URL | Database and Identifier |
|---|---|---|---|---|
| Ahuja S, Whorton MR | 2019 | X-ray crystal structure of the mouse CMP-sialic acid transporter in complex with CMP, by lipidic cubic phase | https://www.rcsb.org/structure/6OH2 | Protein Data Bank, 6OH2 |
| Ahuja S, Whorton MR | 2019 | X-ray crystal structure of the mouse CMP-sialic acid transporter in complex with CMP- sialic acid, by lipidic cubic phase | https://www.rcsb.org/structure/6OH3 | Protein Data Bank, 6OH3 |
| Ahuja S, Whorton MR | 2019 | X-ray crystal structure of the mouse CMP-sialic acid transporter in complex with CMP, by hanging drop vapor diffusion | https://www.rcsb.org/structure/6OH4 | Protein Data Bank, 6OH4 |

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
