## [Decision Letter]

Thank you for submitting your article "Structural basis for mammalian nucleotide sugar transport" for consideration by *eLife*. Your article has been reviewed by three peer reviewers, one of whom is a member of our Board of Reviewing Editors, and the evaluation has been overseen by Richard Aldrich as the Senior Editor. The following individuals involved in review of your submission have agreed to reveal their identity: Ming Zhou (Reviewer #2); Gary Rudnick (Reviewer #3).

The reviewers have discussed the reviews with one another and the Reviewing Editor has drafted this decision to help you prepare a revised submission.

Summary:

The manuscript titled "Structural basis for mammalian nucleotide sugar transport" by Ahuja and Whorton reports crystallization and structure determination of the mouse CMP-sialic acid (CMP-Sia) transporter. The purified mCST protein was shown to bind substrate CMP-Sia and CMP with the expected affinities and to transport CMP-Sia when reconstituted into liposomes. Substantial transport of CMP-Sia happens only when there is a concentration gradient of CMP in the opposite direction of CMP-Sia uptake, which is consistent with the definition that mCST is a CPM-sialic acid-CMP antiporter. The authors then described substrate recognition and identified several residues that are important for substrate selectivity. Finally, the authors compared the current structure with the previously solved transporter structures of the same fold belonging to the drug/metabolite (DMT) superfamily. They identified relative movements between helices 1 and 9 as required for substrate transport, and deduced that helices 5, 9, and 10 would move as a rigid body to open or close the binding site to the extracellular (lumen) side. They further postulated that helices 2, 7, and 8 would move as a rigid body to open and close the binding site to the intracellular (cytosolic) side. Overall, all reviewers have appreciated the high quality of the experimental work. There was, however, a consensus opinion that the proposed models might over-interpret the data. It was concluded that the authors should revise the manuscript to tone down the mechanistic interpretations.

Essential revisions:

1) It is not disputed that differences in homologous structures could provide hints on conformation changes relevant to substrate translocation. However, the conclusion that TMs 5, 9, and 10 move as a rigid body is not substantiated by any known experiments. Fluorescence change of W207 does not provide enough support for the proposed movement. The authors speculate that opening up on the cytoplasmic side involves similar conformational changes in the pseudo-symmetrical repeats. This is a reasonable hypothesis, but it might be prudent to mention that a pseudo-symmetrical structure with inverted repeats doesn't guarantee a symmetrical mechanism. LeuT is an example of a transporter for which the conformational changes that open the extracellular and cytoplasmic pathways are quite different despite pseudo-symmetry. Furthermore, it does not seem right to make conclusions about how one conformational change causes the other (subsection “Transport mechanism”, third paragraph) without experimental or computational data to support the speculation or without at least observed changes in specific interactions (e.g., formation or disruption of hydrogen bonds). Similarly, in the fourth paragraph of the aforementioned subsection, it is unclear how do the authors know that interactions between TM1 and 9 are weaker? Again, implying causality in the last paragraph of the aforementioned subsection seems to be overreaching. The authors should recast their mechanistic interpretations in less definitive terms, discuss possible alternatives, and compare to mechanisms previously proposed for other DMT proteins.

2) While reviewers agree with the assignment of H-bonds between cytidine and the protein, they are not certain that one could attribute entirely the preference for CMP over UMP to difficulty of forming hydrogen bond between uridine and residues N210 and Y214. The authors could look into either electron distribution or the electrostatic potentials of the two nucleobase rings and if the property is substantially different, then match the difference with the electrostatic potential of the binding pocket.

3) The authors speculate on the importance of this intermediate in the mechanism of coupling CMP-Sia into the lumen in exchange for CMP efflux out to the cytoplasm. Fundamentally, the coupling mechanism for an exchanger like this is the inability of the transporter to convert between inward-open and outward-open (cytoplasmic and lumenal) unless one substrate or the other is bound. The energetics of the process are controlled by the relative transmembrane concentration differences of the two substrates. However, the authors seem to want to describe the coupling in terms of affinity of substrates to the various structures. These affinities can act to enhance the relative rate of binding and dissociation, but they do not control the coupling. The most important thing that binding can do for coupling is for the substrate-transporter interactions to overcome a high energy barrier that prevents the conversion between inward- and outward-facing conformations. However, if substrate binding to the semi-occluded intermediate presented here were too strong, it would slow down exchange by tying up all transporters in this state.

4) It is difficult to deduce why CMP-Sia binds weaker than CMP alone. It seems that subtle – on the order of 0.5 Å – movements of the side chains involved in coordination of CMP moiety may not be the key or at least not the only important factors. The overall protein distortion necessary to accommodate the sugar moiety might itself be energetically costly. Other factors cannot be excluded either. These points should be clarified in the text (subsection “Comparison of the CMP and CMP-Sia structures”).

5) It may not be appropriate to say that movement of TM 1 leads to increased interactions and reorientation of other helices (subsection “Comparison of the CMP and CMP-Sia structures”, last paragraph). The strength of interactions has not been evaluated, and causality in conformational change has not been established. All that can be said based on the structures alone is that there is a concerted change observed.

---

## [Author Response]

Essential revisions:1) It is not disputed that differences in homologous structures could provide hints on conformation changes relevant to substrate translocation. However, the conclusion that TMs 5, 9, and 10 move as a rigid body is not substantiated by any known experiments.

We have removed claims that TMs 5, 9, and 10 move as a rigid body. We now simply propose that there would be a rearrangement of these TMs (subsection “Alternating access transport model”, second paragraph.

Fluorescence change of W207 does not provide enough support for the proposed movement.

Our data on the fluorescence change of W207 was not meant to support the rigid body motion of TMs 5, 9, and 10. We have now clarified the text to say that the data on the fluorescence change of W207 merely support the concept that mCST may transition between a substrate-free fully-open state and a substrate-bound partially-occluded state, which would involve the movement of TMs 1, 2, and 8. We acknowledge that these data do not definitively prove that such a conformational change would occur. We are careful to only state that the data are merely consistent with such a conformational change occurring (subsection “Mechanism of substrate accessibility between mCST and the Golgi lumen”, last paragraph).

The authors speculate that opening up on the cytoplasmic side involves similar conformational changes in the pseudo-symmetrical repeats. This is a reasonable hypothesis, but it might be prudent to mention that a pseudo-symmetrical structure with inverted repeats doesn't guarantee a symmetrical mechanism. LeuT is an example of a transporter for which the conformational changes that open the extracellular and cytoplasmic pathways are quite different despite pseudo-symmetry.

Thank you for pointing this out. We now include in our Discussion that having pseudo-symmetry does not guarantee a symmetrical exchange mechanism (subsection “Alternating access transport model”, first paragraph.

Furthermore, it does not seem right to make conclusions about how one conformational change causes the other (subsection “Transport mechanism”, third paragraph) without experimental or computational data to support the speculation or without at least observed changes in specific interactions (e.g., formation or disruption of hydrogen bonds).

We agree with this point. In our restructured discussion of the transport mechanisms, we now no longer make conclusions about the causality of conformational changes.

Similarly, in the fourth paragraph of the aforementioned subsection, it is unclear how do the authors know that interactions between TM1 and 9 are weaker?

Thank you for pointing this out. We have removed claims about relative strengths of interactions. We now describe the TM interactions in terms of buried surface area and discuss how this suggests that one conformation may be more stable than another (subsection “Structural basis for obligatory antiport”, fourth paragraph).

Again, implying causality in the last paragraph of the aforementioned subsection seems to be overreaching.

We agree and have removed these claims of causality.

The authors should recast their mechanistic interpretations in less definitive terms, discuss possible alternatives, and compare to mechanisms previously proposed for other DMT proteins.

In our restructured discussion of the transport mechanisms, we describe things in less definitive terms and have included discussion of possible alternatives as well as comparisons to mechanisms proposed for other DMT proteins.

2) While reviewers agree with the assignment of H-bonds between cytidine and the protein, they are not certain that one could attribute entirely the preference for CMP over UMP to difficulty of forming hydrogen bond between uridine and residues N210 and Y214. The authors could look into either electron distribution or the electrostatic potentials of the two nucleobase rings and if the property is substantially different, then match the difference with the electrostatic potential of the binding pocket.

Thank you for this suggestion. We have performed this analysis and while we did not notice any prominent electrostatic features around the nucleobase, we now discuss several other mechanisms that might contribute to CMP/UMP selectivity (subsection “Nucleotide substrate recognition and selectivity”, second paragraph).

3) The authors speculate on the importance of this intermediate in the mechanism of coupling CMP-Sia into the lumen in exchange for CMP efflux out to the cytoplasm. Fundamentally, the coupling mechanism for an exchanger like this is the inability of the transporter to convert between inward-open and outward-open (cytoplasmic and lumenal) unless one substrate or the other is bound. The energetics of the process are controlled by the relative transmembrane concentration differences of the two substrates. However, the authors seem to want to describe the coupling in terms of affinity of substrates to the various structures. These affinities can act to enhance the relative rate of binding and dissociation, but they do not control the coupling. The most important thing that binding can do for coupling is for the substrate-transporter interactions to overcome a high energy barrier that prevents the conversion between inward- and outward-facing conformations. However, if substrate binding to the semi-occluded intermediate presented here were too strong, it would slow down exchange by tying up all transporters in this state.

We agree with these points. The main point we are trying to make is that one reason CMP may be more effective at trans-stimulation is that for a given concentration it will have a higher fractional occupancy of the partially-occluded state. We have rephrased this argument and it is hopefully clearer now (subsection “Structural basis for obligatory antiport”, last paragraph).

4) It is difficult to deduce why CMP-Sia binds weaker than CMP alone. It seems that subtle – on the order of 0.5 Å – movements of the side chains involved in coordination of CMP moiety may not be the key or at least not the only important factors. The overall protein distortion necessary to accommodate the sugar moiety might itself be energetically costly. Other factors cannot be excluded either. These points should be clarified in the text (subsection “Comparison of the CMP and CMP-Sia structures”).

Thank you for this suggestion. We have now expanded our Discussion to include several factors that may contribute to the differential binding affinity between CMP and CMP-Sia, including potential energetic costs as well as the role of the charge density on the phosphate (subsection “Structural basis for the differential binding affinities of CMP and CMP-Sia”, last two paragraphs).

5) It may not be appropriate to say that movement of TM 1 leads to increased interactions and reorientation of other helices (subsection “Comparison of the CMP and CMP-Sia structures”, last paragraph). The strength of interactions has not been evaluated, and causality in conformational change has not been established. All that can be said based on the structures alone is that there is a concerted change observed.

Thank you for pointing this out. We agree and have removed claims of causality.